# CROSS-DIMENSIONAL SELF-ATTENTION FOR MULTIVARIATE, GEO-TAGGED TIME SERIES IMPUTATION

## ABSTRACT

Many real-world applications involve multivariate, geo-tagged time series data: at each location, multiple sensors record corresponding measurements. For example, air quality monitoring system records PM2.5, CO, etc. The resulting time-series data often has missing values due to device outages or communication errors. In order to impute the missing values, state-of-the-art methods are built on Recurrent Neural Networks (RNN), which process each time stamp sequentially, prohibiting the direct modeling of the relationship between distant time stamps. Recently, the self-attention mechanism has been proposed for sequence modeling tasks such as machine translation, significantly outperforming RNN because the relationship between each two time stamps can be modeled explicitly. In this paper, we are the first to adapt the self-attention mechanism for multivariate, geo-tagged time series data. In order to jointly capture the self-attention across different dimensions (i.e. time, location and sensor measurements) while keep the size of attention maps reasonable, we propose a novel approach called Cross-Dimensional Self-Attention (**CDSA**) to process each dimension sequentially, yet in an order-independent manner. On three real-world datasets, including one our newly collected NYC-traffic dataset, extensive experiments demonstrate the superiority of our approach compared to state-of-the-art methods for both imputation and forecasting tasks.

## 1 INTRODUCTION

Various monitoring applications, such as those for air quality (Zheng et al. (2015)), health-care (Silva et al. (2012)) and traffic (Jagadish et al. (2014)), widely use networked observation stations to record multivariate, geo-tagged time series data. For example, air quality monitoring systems employ a collection of observation stations at different **locations**; at each location, multiple sensors concurrently record different **measurements** such as PM2.5 and CO over **time**. Such time series are important for advanced investigation and also are useful for future forecasting. However, due to unexpected sensor damages or communication errors, missing data is unavoidable. It is very challenging to impute the missing data because of the diversity of the missing patterns: sometimes almost random while sometimes following various characteristics.

Traditional data imputation methods usually suffer from imposing strong statistical assumptions. For example, Scharf & Demeure (1991) and Friedman et al. (2001) fit a *smooth curve* on observations in either time series (Ansley & Kohn (1984); Shumway & Stoffer (1982)) or spatial distribution (Friedman et al. (2001); Stein (2012)). Deep learning methods (Li et al. (2018); Che et al. (2018); Cao et al. (2018); Luo et al. (2018a)) have been proposed to capture temporal relationship based on RNN (Cho et al. (2014b); Hochreiter & Schmidhuber (1997); Cho et al. (2014a)). However, due to the constraint of sequential computation over time, the training of RNN cannot be parallelized and thus is usually time-consuming. Moreover, the relationship between each two distant time stamps cannot be directly modeled. Recently, the self-attention mechanism as shown in Fig. 1(b) has been proposed by the seminal work of *Transformer* (Vaswani et al. (2017)) to get rid of the limitation of sequential processing, accelerating the training time substantially and improving the performance significantly on seq-to-seq tasks in Natural Language Processing (NLP) because the relevance between each two time stamps is captured explicitly.

In this paper, we are the first to adapt the self-attention mechanism to impute missing data in multivariate time series, which cover multiple geo-locations and contain multiple measurements as

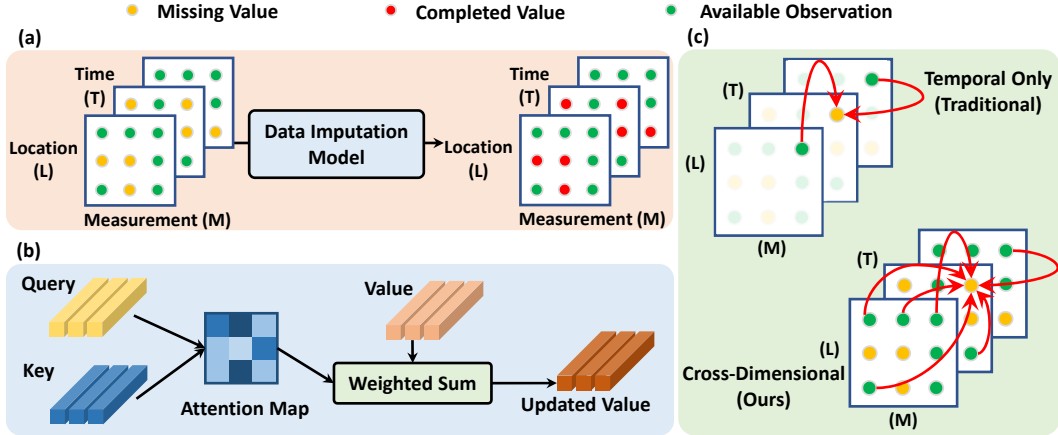

Figure 1: (a) **Illustration of the multivariate, geo-tagged time series imputation task**: the input data has three dimensions (i.e. time, location, measurement) with some missing values (indicated by the orange dot); the output is of same shape as the input while the missing values have been imputed (indicated by the red dot). (b) **Self-attention mechanism**: the **Attention Map** is first computed using every pair of **Query** vector and **Key** vector and then guides the updating of **Value** vectors via weighted sum to take into account contextual information. (c) **Traditional Self-Attention** mechanism updates Value vector along the temporal dimension only vs. **Cross-Dimensional Self-Attention** mechanism updates Value vector according to data across all dimensions.

shown in Fig. 1(a). In order to impute a missing value in such unique multi-dimensional data, it is very useful to look into available data in different dimensions (i.e. **time**, **location** and **measurement**), as shown in Fig. 1(c), to capture the intra-correlation individually. To this end, we investigate several choices of modeling self-attention across different dimensions. In particular, we propose a novel Cross-Dimensional Self-Attention (**CDSA**) mechanism to capture the attention crossing all dimension jointly yet in a decomposed manner. In summary, we make the following contributions:

(i) We are the first to apply the self-attention mechanism to the multivariate, geo-tagged time series data imputation task, replacing the conventional RNN-based models to speed up training and directly model the relationship between each two data values in the input data.

(ii) For such unique time series data of multiple dimensions (i.e. **time**, **location**, **measurement**), we comprehensively study several choices of modeling self-attention crossing different dimensions. Our proposed CDSA mechanism models self-attention crossing all dimensions jointly yet in a dimension-wise decomposed way, preventing the size of attention maps from being too large to be tractable. We show that CDSA is independent with the order of processing each dimension.

(iii) We extensively evaluate on two standard benchmarks and our newly collected traffic dataset. Experimental results show that our model outperforms the state-of-the-art models for both data imputation and forecasting tasks. We visualize the learned attention weights which validate the capability of CDSA to capture important cross-dimensional relationships.

## 2 RELATED WORK

**Statistical data imputation methods.** Statistical methods (Ansley & Kohn (1984); Zhang (2003); Shumway & Stoffer (1982); Nelwamondo et al. (2007); Buuren & Groothuis-Oudshoorn (2010)) often impose assumptions over data and reconstruct the missed value by fitting a *smooth curve* to the available values. For instance, Kriging variogram model (Stein (2012)) was proposed to capture the variance in data w.r.t. the geodesic distance. Matrix completion methods (Acuna & Rodriguez (2004); Yu et al. (2016); Friedman et al. (2001); Cai et al. (2010); Ji & Ye (2009); Ma et al. (2011)) usually enforce low-rank constraint.

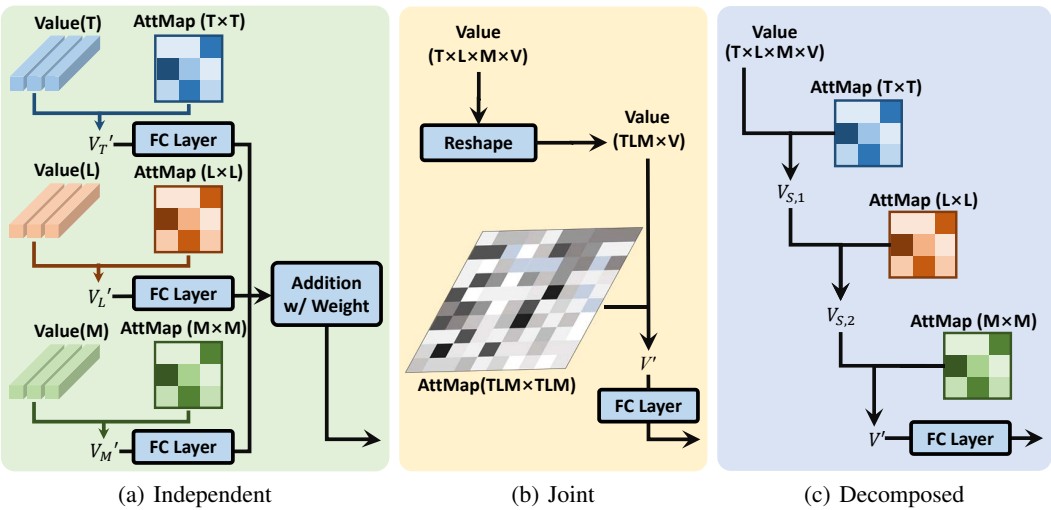

Figure 2: Three choices of implementing our Cross-Dimensional Self-Attention mechanism

**RNN-based data imputation methods.** Li et al. (2018) proposed DCGRU for seq-to-seq by adopting graph convolution (Chung & Graham (1997); Shi (2009); Shuman et al. (2012)) to model spatial-temporal relationship. Luo et al. (2018a) built GRUI by incorporating RNN into a Generative Adversarial Network (GAN). Nevertheless, the spatiotemporal and measurements correlation are mixed and indistinguishable. so that the mediate back propagation from loss of available observation can contribute to the missing value updating. Nevertheless, these RNN-based models fundamentally suffer from the constraint of sequential processing, which leads to long training time and prohibits the direct modeling of the relationship between two distant data values.

**Self-attention.** Recently, Vaswani et al. (2017) introduced the *Transformer* framework which relies on self-attention, learning the association between each two words in a sentence. Then self-attention has been widely applied in seq-to-seq tasks such as machine translation, image generation (Yang et al. (2016); Zhang et al. (2018a)) and graph-structured data (Veličković et al. (2017)). In this paper, we are the first to apply self-attention for multi-dimensional data imputation and specifically we investigate several choices of modeling self-attention crossing different data dimensions.

## 3 APPROACH

In Sec. 3.1, we first review the conventional self-attention mechanism in NLP. In Sec. 3.2, we propose three methods for computing attention map cross different dimension. In Sec. 3.3 and 3.4, we present details of using CDSA for missing data imputation.

### 3.1 CONVENTIONAL SELF-ATTENTION

As shown in Fig. 1(b), for language translation task in NLP, given an input sentence, each word $x_i$ is mapped into a *Query* vector $q_i$ of $d$-dim, a *Key* vector $k_i$ of $d$-dim, and a *Value* vector $v_i$ of $v$-dim. The attention from word $x_j$ to word $x_i$ is effectively the scaled dot-product of $q_i$ and $k_j$ after Softmax, which is defined as $A(i,j) = \exp(S(i,j))\left(\sum_{j=1}^{T} \exp(S(q,j))\right)^{-1}$ where Then, $v_i$ is updated to $v_i'$ as a weighted sum of all the *Value* vectors, defined as $v_i' = \sum_{j=1}^{T} A(i,j)v_j$, after which each $v_i'$ is mapped to the layer output $x_i'$ of the same size as $x_i$. In order to adapt the self-attention from NLP to our multivariate, geo-tagged time series data, a straightforward way is to view all data in a time stamp as one word embedding and model the self-attention over time.

### 3.2 CROSS-DIMENSIONAL SELF-ATTENTION

In order to model Cross-Dimensional Self-Attention (CDSA), in this section we propose three solutions: (1) model attention within each dimension **independently** and perform late fusion; (2)

model attention crossing all dimension **jointly**; (3) model attention crossing all dimension in a joint yet **decomposed** manner. We assume the input $\mathcal{X} \in \mathbb{R}^{T \times L \times M}$ has three dimensions corresponding time, location, measurement. $\mathcal{X}$ can be reshaped into 2-D matrices (i.e. $\boldsymbol{X}_{\mathcal{T}} \in \mathbb{R}^{T \times LM}$, $\boldsymbol{X}_{\mathcal{L}} \in \mathbb{R}^{L \times MT}$, $\boldsymbol{X}_{\mathcal{M}} \in \mathbb{R}^{M \times TL}$) or an 1-D vector (i.e. $\boldsymbol{X} \in \mathbb{R}^{TLM \times 1}$). Similarly, this subscript may be applied on the *Query*, *Key* and *Value*, e.g., $\mathcal{Q} \in \mathbb{R}^{T \times L \times M \times d}$, $\boldsymbol{Q}_{\mathcal{L}} \in \mathbb{R}^{L \times MT d}$ and $\boldsymbol{Q} \in \mathbb{R}^{TLM \times d}$.

### 3.2.1 INDEPENDENT

As shown in Fig. 2(a), the input $\mathcal{X}$ is reshaped into three input matrices $\boldsymbol{X}_{\mathcal{T}}$, $\boldsymbol{X}_{\mathcal{L}}$ and $\boldsymbol{X}_{\mathcal{M}}$. Three streams of self-attention layers are built to process each input matrix in parallel. Such as the first layer in stream on $\boldsymbol{X}_{\mathcal{L}}$, each vector $\boldsymbol{X}_{\mathcal{L}}(l,:)$ of $MT$-dim is viewed as a word vector in NLP. Following the steps in Sec. 3.1, $\boldsymbol{X}_{\mathcal{L}}(l,:)$ is mapped to $\boldsymbol{Q}_L(l,:)$ and $\boldsymbol{K}_L(l,:)$ of $d_L$-dim, as well as $\boldsymbol{V}_L(l,:)$ of $v_L$-dim. The output of every stream's last layer are fused through element-wise addition, $\mathcal{X}' = \alpha_T \mathcal{X}'_T + \alpha_L \mathcal{X}'_L + \alpha_M \mathcal{X}'_M$, where the weights $\alpha_T$, $\alpha_L$ and $\alpha_M$ are trainable parameters. Besides, the hyper-parameters for each stream such as the number of layers, are set separately.

### 3.2.2 JOINT

As shown in Fig. 2(b), the three-dimensional input $\mathcal{X}$ is reshaped as to $\boldsymbol{X}$. Each unit $\boldsymbol{X}(p)$ is mapped to $\boldsymbol{Q}(p,:)$ and $\boldsymbol{K}(p,:)$ of $d$-dim as well as $\boldsymbol{V}(p,:)$ of $v$-dim, where $p = p(t,l,m)$ denotes the index mapping from the 3-D cube to the vector form. In this way, an attention map of dimension $TLM \times TLM$ is built to directly model the cross-dimensional interconnection.

### 3.2.3 DECOMPOSED

The *Independent* manner sets multiple attention sub-layers in each stream to model the dimension-specific attention but fail in modeling cross-dimensional dependency. In contrast, the *Joint* manner learns the cross-dimensional relationship between units directly but results in huge computation workload. To capture both the dimension-specific and cross-dimensional attention in a distinguishable way, we propose a novel *Decomposed* manner.

As shown in Fig. 2(c), the input $\mathcal{X}$ is reshaped as input matrices $\boldsymbol{X}_{\mathcal{T}}$, $\boldsymbol{X}_{\mathcal{L}}$, $\boldsymbol{X}_{\mathcal{M}}$ and $\boldsymbol{X}$. Each unit $\boldsymbol{X}(p)$ is mapped to vector $\boldsymbol{V}(p,:)$ of $v$-dim as in the *Joint* while $\boldsymbol{X}_{\mathcal{T}}$, $\boldsymbol{X}_{\mathcal{L}}$ and $\boldsymbol{X}_{\mathcal{M}}$ are used for building attention map $\boldsymbol{A}_T$, $\boldsymbol{A}_L$, $\boldsymbol{A}_M$ individually as in the *Independent*. The attention maps are applied on *Value* vector in order as,

$$\boldsymbol{V}' = \boldsymbol{A}\boldsymbol{V} = \widetilde{\boldsymbol{A}}_M \boldsymbol{V}_{\text{S},2} = \widetilde{\boldsymbol{A}}_M \widetilde{\boldsymbol{A}}_L \boldsymbol{V}_{\text{S},1} = \widetilde{\boldsymbol{A}}_M \widetilde{\boldsymbol{A}}_L \widetilde{\boldsymbol{A}}_T \boldsymbol{V}. \tag{1}$$

The attention map with $\widetilde{\phantom{A}}$ is reshaped from the original attention map and consistent with the calculation in (1), e.g., $\widetilde{\boldsymbol{A}}_T \in \mathbb{R}^{TLM \times TLM}$ is reshaped from $\boldsymbol{A}_T \in \mathbb{R}^{T \times T}$. More specifically,

$$\begin{aligned}
\widetilde{\boldsymbol{A}}_T &= \boldsymbol{A}_T \otimes \boldsymbol{I}_L \otimes \boldsymbol{I}_M, \\
\widetilde{\boldsymbol{A}}_L &= \boldsymbol{I}_T \otimes \boldsymbol{A}_L \otimes \boldsymbol{I}_M, \\
\widetilde{\boldsymbol{A}}_M &= \boldsymbol{I}_T \otimes \boldsymbol{I}_L \otimes \boldsymbol{A}_M,
\end{aligned} \tag{2}$$

where $\otimes$ denotes *tensor product* and $\boldsymbol{I}$ is the *Identity* matrix where the subscript indicates the size, e.g., $\boldsymbol{I}_T \in \mathbb{R}^{T \times T}$. Although the three reshaped attention maps are applied with a certain order, according to (2), we show that each unit in $\boldsymbol{A}$ is effectively calculated as

$$\boldsymbol{A}(p_0, p_1) = \boldsymbol{A}_T(t_0, t_1) \boldsymbol{A}_L(l_0, l_1) \boldsymbol{A}_M(m_0, m_1), \tag{3}$$

where $p_0 = p(t_0, l_0, m_0), p_1 = p(t_1, l_1, m_1)$. Following the associativity of tensor product, we demonstrate

$$\widetilde{\boldsymbol{A}}_{\sigma(T)} \widetilde{\boldsymbol{A}}_{\sigma(L)} \widetilde{\boldsymbol{A}}_{\sigma(M)} = \boldsymbol{A}_T \otimes \boldsymbol{A}_L \otimes \boldsymbol{A}_M, \tag{4}$$

where $\sigma = \sigma(\text{T,L,M})$ denotes the arbitrary arrangement of sequence (T,L,M), e.g., $\sigma =$(L,T,M) and $\sigma(\text{T}) = \text{L}$. Effectively, the arrangement $\sigma$ is the order of attention maps to update $\boldsymbol{V}$. As (3)-(4) shows that the weight in $\boldsymbol{A}$ is decomposed as the product of weights in three dimension-specific attention maps, the output and gradient back propagation are order-independent. Furthermore, we show in Supp. B that the cross-dimensional attention map has the following property:

$$\sum_{p_1=1}^{TLM} \boldsymbol{A}(p_0, p_1) = \sum_{t_1=1}^{T} \sum_{l_1=1}^{L} \sum_{m_1=1}^{M} \boldsymbol{A}_T(t_0, t_1) \boldsymbol{A}_L(l_0, l_1) \boldsymbol{A}_M(m_0, m_1) = 1. \tag{5}$$

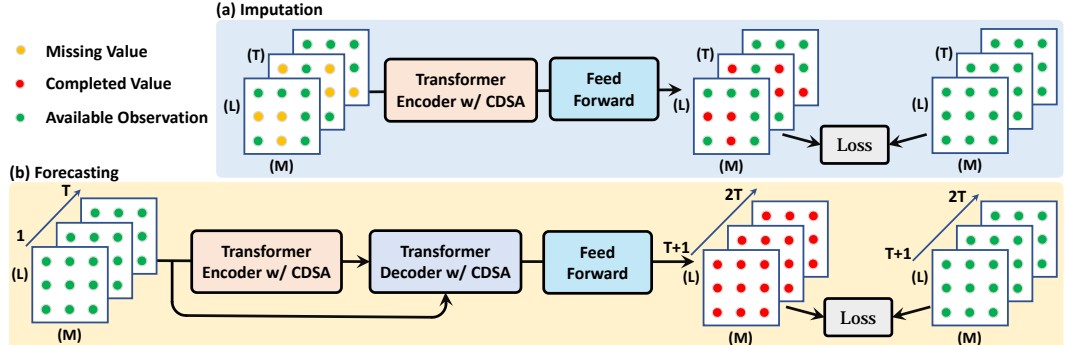

Figure 3: The framework employing CDSA for data imputation and forecasting.

In summary, the *Independent* builds attention stream for each dimension while the *Joint* directly model the attention map among all the units. Our proposed CDSA is based on the *Decomposed*, which forms a cross-dimensional attention map, out of three dimension-specific maps. As an alternative of the *Decomposed*, the *Shared* maps unit $\boldsymbol{X}(p)$ to $\boldsymbol{Q}(p,:)$ and $\boldsymbol{K}(p,:)$ of $d$-dim and calculates all three dimension-specific attention map, e.g., $\boldsymbol{A}_L = \text{Softmax}(\boldsymbol{Q}_{\mathcal{L}}\boldsymbol{K}_{\mathcal{L}}^{\top}/\sqrt{MTd})$. As shown in Table 1, by using Tensorflow *profile* and fixing the hyper-parameters with detailed explanations in Supp., the *Decomposed* significantly decreases the FLoating point OPerations (FLOPs) compared to the *Joint* and requires less variables than the *Independent*. Detailed comparisons are reported in Sec. 4.3.

Table 1: Computational complexity of several methods to implement CDSA

| Methods | *Independent* | *Joint* | *Shared* | *Decomposed* |
|---|---|---|---|---|
| FLOPs($\times 10^9$) | 0.39 | 3.22 | 0.21 | 0.24 |
| Number of Variables ($\times 10^5$) | 18.15 | 0.44 | 0.44 | 16.09 |

### 3.3 FRAMEWORK

**Imputation**: As shown in Fig. 3(a), we apply our CDSA mechanism in a *Transformer **Encoder***, a stack of $N = 8$ identical layers with residual connection (He et al. (2016)) and normalization (Lei Ba et al. (2016)) as employed by Vaswani et al. (2017). To reconstruct the missing (along with other) values of the input, we apply a fully connected *Feed Forward* network on the final Value tensor, which is trained jointly with the rest of the model.

**Forecasting**: As shown in Fig. 3(b), we apply our CDSA mechanism in *Transformer* framework where we set $N = 9$ for both encoder and decoder. Similar to imputation, we use a fully connected feed forward network to generate the predicted values.

### 3.4 IMPLEMENTATION DETAILS

We normalize each measurement of the input by subtracting the mean and dividing by standard deviation across training data. Then the entries with missed value are set 0. We use the Adam optimizer (Kingma & Ba (2014)) to minimize the Root Mean Square Error (RMSE) between the prediction and ground truth. The model is trained on a single NVIDIA GTX 1080 Ti GPU. More details (e.g., network hyper-parameters, learning rate and batch size) can be found in Supp.

## 4 EXPERIMENTS

### 4.1 DATASETS, TASKS, EVALUATION METRICS

**NYC-Traffic**. New York City Department of Transportation has set up various street cameras[1]. Each camera keeps taking a snapshot every a few seconds. The is collected around 1-month data

---

[1]https://nyctmc.org/

for 186 cameras on Manhattan from 12/03/2015 to 12/26/2015. For each snapshot, we apply our trained faster-RCNN (Ren et al. (2015)) vehicle detection model to detect the number of vehicles (#vehicle) in each snapshot. To aggregate such raw data into time series, for every non-overlapping 5-minute window, we averaged #vehicle from each snapshot to obtain the average #vehicle as the only measurement. Finally, we obtained 186 time series and the gap between two consecutive time stamps is 5 minutes.

The natural missing rate of the whole dataset is 8.43%. In order to simulate experiments for imputation, we further remove some entries and hold them as ground truth for evaluation. The imputation task is to estimate values of these removed entries. To mimic the natural data missing pattern, we model our manual removal as a *Burst Loss*, which means at certain location the data is continuously missed for a certain period. More details about vehicle detection and burst loss are be found in Supp. To simulate various data missing extents, we vary the final missing rate after removal from 20% to 90%. For each missing rate, we randomly select 432 consecutive time slots to train our model and evaluate the average RMSE of 5 trials. The dataset will be released publicly.

**KDD-2018** (Cup (2018)) is an Air Quality and Meteorology dataset recorded hourly. As indicated in Luo et al. (2018a), 11 locations and 12 measurements are selected. The natural missing rate is 6.83%. In order to simulate experiments for imputation, we follow Luo et al. (2018a) to split the data to every 48 hours, randomly hold values of some available entries and vary the missing rate from 20% to 90%. Mean Squared Error (MSE) is used for evaluation.

**METR-LA** (Jagadish et al. (2014)). We follow Li et al. (2018) to use this dataset for traffic speed forecasting. This dataset contains traffic speed at 207 locations recorded every 5 minutes for 4 months. Following Li et al. (2018), 80% of data at the beginning of these 4 months is used for training and the remaining 20% is for testing. In order to simulate the forecasting scenario, within either training or testing set, every time series of consecutive 2 hours are enumerated. For each time series, data in the first hour is treated as input and data in the second hour is to be predicted. We respectively evaluate the forecasting results at 15-th, 30-th, 60-th minutes in the second 1 hour and also evalaute the average evaluation results within the total 1 hour. We use RMSE, Mean Absolute Error (MAE) and Mean Absolute Percentage Error (MAPE) as evaluation metrics.

## 4.2 COMPARISONS WITH STATE-OF-THE-ART

Table 2: RMSE on dataset **NYC-Traffic** for comparisons with SOTA

| Model \ Missing Rate | 20% | 30% | 40% | 50% | 60% | 70% | 80% | 90% |
|---|---|---|---|---|---|---|---|---|
| Auto Regressive | 2.354 | 2.357 | 2.359 | 2.362 | 2.364 | 2.652 | 2.796 | 3.272 |
| Kriging expo | 2.142 | 2.145 | 2.157 | 2.152 | 2.155 | 2.165 | 2.182 | 2.231 |
| Kriging linear | 2.036 | 2.008 | 2.031 | 2.038 | 2.056 | 2.074 | 2.111 | 2.194 |
| MTSI Luo et al. (2018a) | 1.595 | 1.597 | 1.603 | 1.605 | 1.608 | 1.641 | 1.672 | 1.834 |
| BRITS Cao et al. (2018) | 1.337 | 1.339 | 1.341 | 1.355 | 1.376 | 1.395 | 1.408 | 1.477 |
| DCRNN Li et al. (2018) | 1.397 | 1.399 | 1.401 | 1.419 | 1.432 | 1.443 | 1.459 | 1.601 |
| **CDSA (ours)** | **1.203** | **1.208** | **1.211** | **1.214** | **1.215** | **1.217** | **1.234** | **1.377** |

Table 3: MSE on dataset **KDD-2018** for comparisons with SOTA

| Model \ Missing Rate | 20% | 30% | 40% | 50% | 60% | 70% | 80% | 90% |
|---|---|---|---|---|---|---|---|---|
| Mean Filling | 0.916 | 0.907 | 0.914 | 0.923 | 0.973 | 0.935 | 0.937 | 1.002 |
| KNN Filling | 0.892 | 0.803 | 0.776 | 0.798 | 0.856 | 0.852 | 0.873 | 1.243 |
| MF Filling | 0.850 | 0.785 | 0.787 | 0.772 | 0.834 | 0.805 | 0.860 | 1.196 |
| MTSI Luo et al. (2018a) | 0.844 | 0.780 | 0.753 | 0.743 | 0.803 | 0.780 | 0.837 | 1.018 |
| BRITS Cao et al. (2018) | 0.455 | 0.421 | 0.372 | 0.409 | 0.440 | 0.482 | 0.648 | 0.725 |
| DCRNN Li et al. (2018) | 0.579 | 0.565 | 0.449 | 0.506 | 0.589 | 0.622 | 0.720 | 0.861 |
| **CDSA (ours)** | **0.373** | **0.393** | **0.287** | **0.291** | **0.387** | **0.495** | **0.521** | **0.631** |

**Imputation** (**NYC-Traffic**, **KDD-2018**) In Table 2 , our CDSA consistently outperforms traditional methods (i.e., Auto Regressive, Kriging expo, Kriging linear) and recent RNN-based methods (i.e. MTSI, BRITS, DCRNN) over a wide range of missing rate. Because CDSA leverages the self-attention mechanism to avoid sequential processing of RNN and directly model the relationship

between distant data. Table 3 shows that our method again achieves significant improvements on cross-dimensional data imputation task. Detailed overview of baselines can be found in Supp.

**Forecasting (METR-LA)**. Table 4 shows that for the forecasting task, our CDSA method outperforms previous methods in most cases. In particular, our method demonstrates clear improvement at long-term forecasting such as 60 min. This again confirms that our CDSA cthe effectiveness of directly modeling the relationship between every two data values (could from different dimensions and of far distance). But *RNN-based methods* and *methods that sequentially conduct spatial conv and temporal conv* fail to model the distant spatio-temporal relationship explicitly.

Table 4: MAE/RMSE/MAPE on dataset **METR-LA** for comparisons with SOTA

| Model | 15min | | | 30min | | |
|---|---|---|---|---|---|---|
| | MAE | RMSE | MAPE | MAE | RMSE | MAPE |
| FC-LSTM Sutskever et al. (2014) | 3.44 | 6.3 | 9.6% | 3.77 | 7.23 | 10.9% |
| MTSI Luo et al. (2018a) | 3.75 | 7.31 | 10.52% | 3.89 | 7.73 | 11.04% |
| BRITS Cao et al. (2018) | 2.86 | 5.46 | 7.49% | 3.37 | 6.78 | 9.13% |
| DCRNN Li et al. (2018) | 2.77 | 5.38 | 7.3% | 3.15 | 6.45 | 8.8% |
| DST-GCNN Wang et al. (2018) | 2.68 | 5.35 | 7.2% | **3.01** | 6.23 | 8.52% |
| GaAN Zhang et al. (2018b) | **2.71** | 5.24 | **6.99%** | 3.12 | 6.36 | 8.56% |
| **CDSA(ours)** | 3.01 | **5.08** | 7.82% | 3.14 | **5.38** | **8.30%** |
| Model | 60min | | | Mean | | |
| | MAE | RMSE | MAPE | MAE | RMSE | MAPE |
| FC-LSTM Sutskever et al. (2014) | 4.37 | 6.89 | 13.2% | 3.86 | 7.41 | 11.2% |
| MTSI Luo et al. (2018a) | 4.22 | 8.39 | 12.15% | 4.01 | 7.59 | 10.85% |
| BRITS Cao et al. (2018) | 3.65 | 7.66 | 10.55% | 3.32 | 6.96 | 9.47% |
| DCRNN Li et al. (2018) | 3.60 | 7.59 | 10.50% | 3.28 | 6.80 | 8.87% |
| DST-GCNN Wang et al. (2018) | 3.41 | 7.47 | 10.25% | - | - | - |
| GaAN Zhang et al. (2018b) | 3.6 | 7.6 | 10.5% | **3.16** | 6.41 | 8.72% |
| **CDSA(ours)** | **3.40** | **6.27** | **9.76%** | **3.16** | **5.48** | **8.50%** |

## 4.3 DISCUSSIONS

**The effects of different training losses**: For the forecasting task in **METR-LA**, we compare the performance by setting different training loss in Table 5 and we can see the performance with RMSE as loss metric achieves the best performance.

Table 5: Comparisons of different losses in *CDSA* on **METR-LA**

| Time | 30min | Ave | 30min | Ave | 30min | Ave | 30min | Ave |
|---|---|---|---|---|---|---|---|---|
| Metric \ Loss | RMSE | | MSE | | MAE | | (RMSE+MAE)/2 | |
| MAE | **3.14** | **3.16** | 3.43 | 3.41 | 3.28 | 3.33 | 3.21 | 3.25 |
| RMSE | **5.38** | **5.48** | 6.20 | 6.11 | 5.67 | 5.83 | 5.55 | 5.70 |
| MAPE | **8.30%** | **8.50%** | 9.32% | 9.19 | 8.70% | 9.00% | 8.53% | 8.80% |

**Ablation study of different cross-dimensional self-attention manners**: We compare the performance for different solutions in CDSA mechanism on the three datasets listed above. 1) The way of attention modeling determines the computational complexity. As shown in Table 1, since the *Independent* calculates dimension-specific *Value* vectors in parallel, the number of variables and FLOPs are larger than those of the *Decomposed*. As the *Joint* and the *Shared* both share the variables for each dimension, the number of variables is small and basically equals with each other. As the *Joint* builds a huge attention map, its FLOPs is much larger than others. Since the *Decomposed* draws attention maps like the *Independent* but shares *Value* like the *Joint*, it reduces the computational complexity significantly. 2) As shown in Table 6 - 8, we evaluate these methods on three datasets and the *Decomposed* always achieves the best performance thanks to the better learning ability compared to the *Joint* and *Shared*. More discussions can be found in Supp.

**Study of using the imputed time series for forecasting**. On **NYC-Traffic** of missing rate 50%, we impute missing values in historical data (using statistical methods and our CDSA respectively) and

then feed the imputed historical data into traffic prediction model ARIMA. We compare performances in terms of RMSE: *Mean Filling* 1.953, *Kriging expo* 1.681, *Kriging linear* 1.733, *MTSI* 1.675, *DCRNN* 1.666, *BRITS* 1.579, *CDSA* 1.536. This indicate that when using the imputed time series for forecasting, our CDSA can achieve significant gains in the downstream forecasting task as well compared to traditional imputation methods. More details can be found in Supp. E.3.

Table 6: Comparisons of different manners to implement CDSA on dataset **NYC-Traffic**.

| Model \ Missing Rate | 20% | 30% | 40% | 50% | 60% | 70% | 80% | 90% |
|---|---|---|---|---|---|---|---|---|
| CDSA(Independent) | 1.327 | 1.327 | 1.331 | 1.355 | 1.362 | 1.379 | 1.393 | 1.425 |
| CDSA(Joint) | | | Not Applicable due to memory usage limitation | | | | | |
| CDSA(Shared) | 1.637 | 1.645 | 1.651 | 1.657 | 1.684 | 1.729 | 1.733 | 1.935 |
| **CDSA(Decomposed)** | **1.204** | **1.208** | **1.211** | **1.214** | **1.215** | **1.217** | **1.235** | **1.377** |

Table 7: Comparisons of different manners to implement CDSA on dataset **KDD-2018**.

| Model \ Missing Rate | 20% | 30% | 40% | 50% | 60% | 70% | 80% | 90% |
|---|---|---|---|---|---|---|---|---|
| CDSA(Independent) | 0.482 | 0.523 | 0.351 | 0.366 | 0.484 | 0.573 | .608 | 0.721 |
| CDSA(Joint) | 0.451 | 0.497 | 0.317 | 0.336 | 0.404 | 0.520 | 0.558 | 0.677 |
| CDSA(Shared) | 0.783 | 0.799 | 0.672 | 0.692 | 0.784 | 0.793 | 0.791 | 0.832 |
| **CDSA(Decomposed)** | **0.373** | **0.393** | **0.287** | **0.291** | **0.387** | **0.495** | **0.521** | 0.**631** |

Table 8: Comparisons of different manners to implement CDSA on dataset **METR-LA**.

| Model \ Dataset | 60 min | | | Mean | | |
|---|---|---|---|---|---|---|
| | MAE | RMSE | MAPE | MAE | RMSE | MAPE |
| CDSA(Independent) | 3.54 | 7.02 | 10.29% | 3.25 | 6.29 | 8.81% |
| CDSA(Joint) | 3.63 | 7.62 | 10.54% | 3.30 | 6.83 | 9.43% |
| CDSA(Shared) | 3.92 | 7.93 | 11.17% | 3.53 | 7.33 | 10.26% |
| **CDSA(Decomposed)** | **3.40** | **6.27** | **9.76%** | **3.16** | **5.48** | **8.50%** |

**Attention Map Visualization**: Fig. 4 shows an PM10 imputation example in location *fangshan* at $t_2$. Since the pattern of PM2.5 around $t_2$ is similar to that at $t_1$, the attention in orange box is high. As we can see that PM2.5 and PM10 are strongly correlated , in order to impute PM10 at $t_2$, our model utilizes PM10 at $t_1$ (green arrow) and PM2.5 at $t_1$ (blue arrow), which crosses dimensions. More visualization examples can be found in Supp.

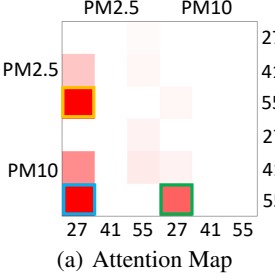

(a) Attention Map

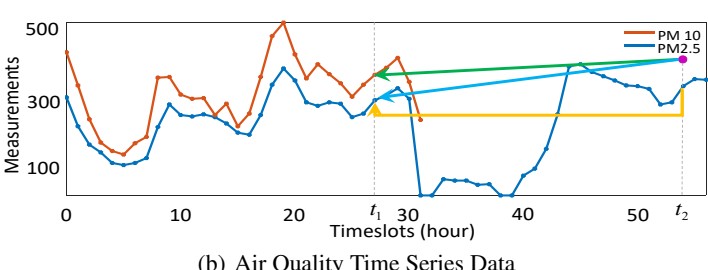

(b) Air Quality Time Series Data

Figure 4: Visualization of the cross-dimensional self-attention on **KDD-2018**. (a) Part of *Time-Measurement* attention map. (b) Two time series of PM2.5 and PM10. The value at purple dot is missing and our model predicts its value based on other values. The arrow in (b) represents attention whose score is highlighted with bounding box in (a) of the same color.

## 5 CONCLUSION

In this paper, we have proposed a cross-dimensional self-attention mechanism to impute the missing values in multivariate, geo-tagged time series data. We have proposed and investigated three methods to model the cross-dimensional self-attention. Experiments show that our proposed model achieves superior results to the state-of-the-art methods on both imputation and forecasting tasks. Given the encouraging results, in the future we plan to extend our CDSA mechanism from multivariate, geo-tagged time series to the input that has higher dimension and involves multiple data modalities.

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

# A    MODEL ARCHITECTURE

## A.1    NORMALIZATION LAYER

Under the NLP scenario, each word is embedded as a vector and normalized individually. However, in the Cross-Dimensional scenario, the normalization applied on each individual unit will always lead to a zero-output. As shown in Fig. 5, different measurements may exhibit different correlation, i.e., PM2.5 and PM 10 are significantly positively correlated ($\rho_{\text{PM2.5, PM 10}} = 0.8278$) while NO$_2$ and O$_3$ are negatively correlated ($\rho_{\text{NO}_2,\text{O}_3} = -0.5117$). As discussed in Fig. 4(b) in the paper, different measurement may be used as reference for imputation of other measurements. As such, the normalization cross multiple measurements is unreasonable and we choose to apply normalization for each measurement in parallel which presumes that the time series inside the spatial network is essentially drawn from a standard normal distribution. We subsequently add the trainable scalar and bias to scale the normalized value and the scalars (biases) for different measurements are trained individually.

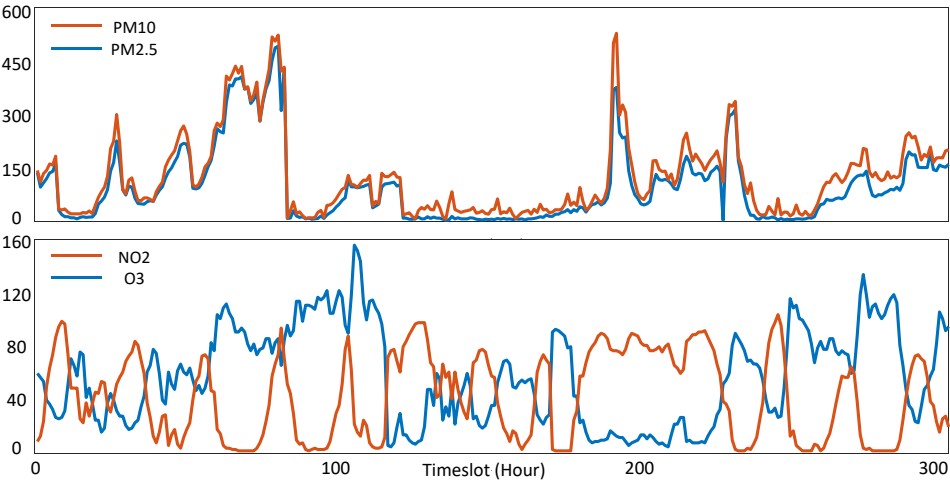

Figure 5: Different correlation between different measurements. The time slots when both of the chosen measurements are available are selected and the value of the first 300 selected time slots are plotted. *Upper*: PM2.5 & PM10 are significantly positively correlated and *Lower*: NO$_2$ & O$_3$ are negatively correlated.

## A.2    UNIT-WISE FEED-FORWARD NETWORK

Making use of the approximation property of multi-linear layer Hornik et al. (1989), a fully connected feed-forward network (FFN) is applied to each unit separately and identically. This FFN consists of three fully connected layer while RuLU is set as the activation function.

$$\text{FFN}(x) = \max(\max(0, \boldsymbol{x}W_1 + b_1)W_2 + b_2)W_3 + b_3 \tag{6}$$

During experiment, since the FFN is applied on each unit individually, we found the improvement by simply increasing the size of weight and bias of each layer is not obvious while increasing the depth of FFN will lead to obvious improvement.

## A.3 IMPUTATION MASK

For the self-attention sub-layer in imputation task **NYC** and **KDD-2018**, we modify the attention map with mask in (7) to prevent unit of available observation from contributing to the estimation of itself.

$$\boldsymbol{S}(i,j) = \begin{cases} -\infty & i = j \\ \boldsymbol{q}_i \boldsymbol{k}_j^\top / \sqrt{d} & \text{otherwise} \end{cases} \tag{7}$$

where $\boldsymbol{q}_i$ and $\boldsymbol{k}_j$ are $d$-dim vectors. This masking, combined with fact that there is no offset between input and output, ensures that the estimation of unit $\boldsymbol{X}(p(t,l,m))$ depends on all the units except for itself, including both available and *complemented* units. In this way, the gradient back-propagation can be used to update the missed value effectively. The Table 9 shows the performance improvement of imputation mask applied in our model and demonstrate that mask prevents the estimation of itself and improve the inference ability of the model.

Table 9: Performance Improvement for Imputation Mask

| Model \ Missing Rate | | 20% | 30% | 40% | 50% | 60% | 70% | 80% | 90% |
|---|---|---|---|---|---|---|---|---|---|
| **KDD-2018** | Masked | 0.373 | 0.393 | 0.287 | 0.291 | 0.387 | 0.495 | 0.521 | 0.631 |
| | No Mask | 0.886 | 0.839 | 0.838 | 0.849 | 0.917 | 0.903 | 0.947 | 1.195 |
| **NYC** | Masked | 1.04 | 1.10 | 1.32 | 1.18 | 1.18 | 1.28 | 1.29 | 1.75 |
| | No Mask | 1.69 | 1.70 | 1.72 | 1.77 | 1.80 | 1.84 | 1.91 | 2.04 |

## A.4 ATTENTION MAP CALCULATION

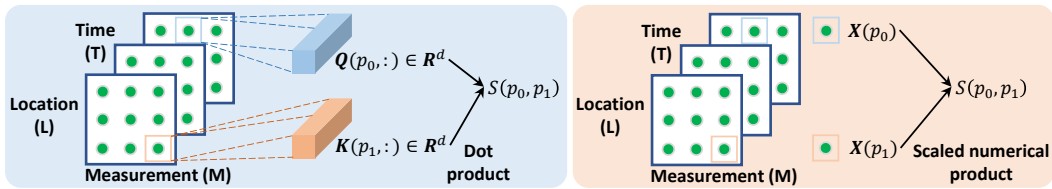

Figure 6: The effective attention map calculation in *Joint* and *Shared*.

**Joint** As shown in Fig. 6, when we build the attention among different units in the *Joint*, two different kernels will be used to map each input unit $\boldsymbol{X}(p) = \mathcal{X}(t,l,m)$ to an 1-D *Query* vector and an 1-D *Key* vector individually. As attention map is a scaled dot-product between *Query* and *Key* (Fig. 6 Left) after Softmax, each value of attention map in *Joint* is essentially the scaled numerical multiplication between each two units of input (Fig. 6 Right) after Softmax. As such, the multiple parameters inside that kernels only perform as a single scalar and the learning ability/relationship representation in *Joint* is limited.

**Decomposed** According to Sec. 3.2.3, to calculate the dimension-specific attention map, e.g., the attention map of *Time* $\boldsymbol{A}_T$, the input $\mathcal{X}$ will be reshaped into matrix $\boldsymbol{X}_\mathcal{T}$. Thus, the units corresponding to the same timestamp, reshaped into one vector $\boldsymbol{X}_\mathcal{T}(t,:)$, will be mapped into dimension-specific *Query* vector $\boldsymbol{Q}_T(t,:)$ and *Key* vector $\boldsymbol{K}_T(t,:)$. As more parameters will be introduced into such vector-vector mapping, each dimensional-specific map can learn the intra-correlation and the cross-dimensional attention map can model the relationship among each input units effectively.

**Shared** Same with the *Decomposed*, the *Shared* will calculate the dimension-specific attention maps individually. Like *Joint*, the *Shared* will map each unit $\boldsymbol{X}(p)$ into vector. To calculate the dimension-specific attention map (e.g., $\boldsymbol{A}_T$), the "vector-vector mapping" is essentially the summation of the units corresponding to the same timestamp while the multiple parameters introduced in the mapping still perform as a single scalar. As a result, the learning ability of the intra-correlation is limited so that the cross-dimensional attention map cannot model the relationship among input units effectively.

# B   ATTENTION MAP RESHAPE IN THE *Decomposed*

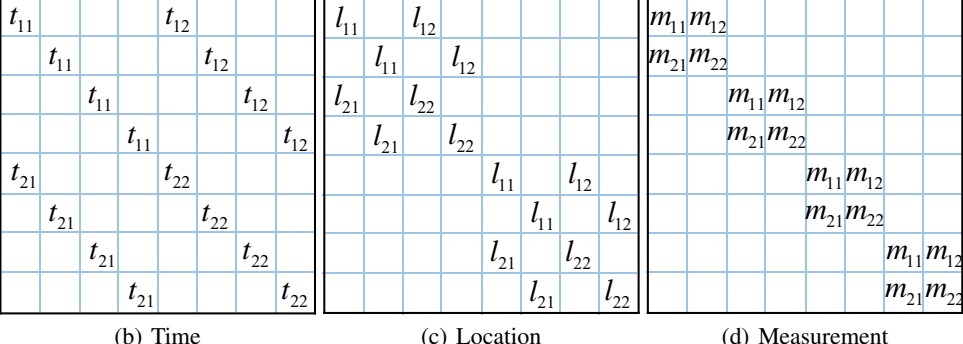

(a) Attention Map of *Time*, *Location* and *Measurement*

(b) Time          (c) Location          (d) Measurement

Figure 7: **Reshape Attention Map**: The original attention map in (a) is reshaped into (b-d) respectively. The variable $(t, l, m)$ denotes the attention map units for *Time*, *Location* and *Measurement* where the subscript labels the attention between two units, i.e., $t_{12} = \text{Softmax}(\boldsymbol{q}_1 \boldsymbol{k}_2^\top / \sqrt{d})$ where $\boldsymbol{q}_1$ and $\boldsymbol{k}_2$ are $d$-dim vectors. Besides, the empty entry indicates *0*.

As described in (2) in the paper, the original attention maps $\boldsymbol{A}_T$, $\boldsymbol{A}_L$ and $\boldsymbol{A}_M$ are reshaped to $\boldsymbol{A}'_T$, $\boldsymbol{A}'_L$ and $\boldsymbol{A}'_M$. By setting $T = L = M = 2$ as an example, we draw the attention maps before and after reshape in Fig. 7. Making use of the matrix structure, we have

$$
\sum_{p_1=1}^{TLM} \boldsymbol{A}(p_0, p_1) = \sum_{t_1=1}^{T} \sum_{l_1=1}^{L} \sum_{m_1=1}^{M} \boldsymbol{A}_T(t_0, t_1) \boldsymbol{A}_L(l_0, l_1) \boldsymbol{A}_M(m_0, m_1)
$$
$$
= \sum_{t_1=1}^{T} \left( \boldsymbol{A}_T(t_0, t_1) \sum_{l_1=1}^{L} \left( \boldsymbol{A}_L(l_0, l_1) \sum_{m_1=1}^{M} \boldsymbol{A}_M(m_0, m_1) \right) \right) = 1,
$$

(8)

where $\sum_{t_1=1}^{T} \boldsymbol{A}_T(t_0, t_1) = 1$, $\sum_{l_1=1}^{L} \boldsymbol{A}_L(l_0, l_1) = 1$ and $\sum_{m_1=1}^{M} \boldsymbol{A}_M(m_0, m_1) = 1$. Besides, the reshape operation for attention map on different dimension is only determined by the index mapping function, $p = p(t, l, m) = LMt + Ml + m$, from the 3-dimensional cube to the vector form.

## C  CDSA FRAMEWORK FOR DATA FORECASTING

Different from time series imputation task, the time series forecasting task use the current observation to estimate the time series in the future. To begin with, We first introduce the framework for time series forecasting and we compare the performance for two different types of input.

### C.1  PREDICTION FRAMEWORK

As shown in Fig. 8, we apply our CDSA mechanism in *Transformer* framework and use the same *Feed Forward* structure as in Sec. 3.3 in the paper. Notably, we set $N = 9$ for both encoder and decoder and no CDSA module is used to derive a *complement* input for prediction task where the missing value is replaced with global mean. The architecture detail is shown in Fig. 9

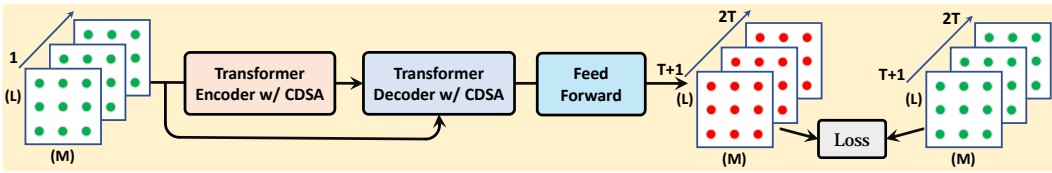

Figure 8: The framework of using Crossing-Dimensional Self-Attention (CDSA) for data forecasting.

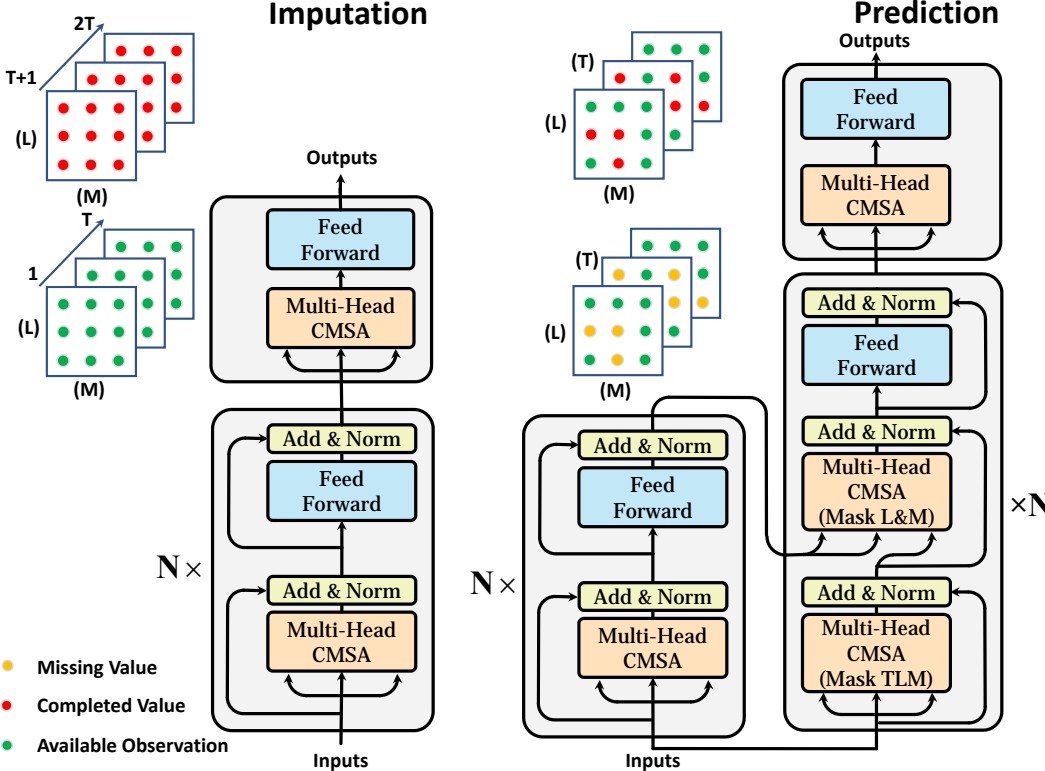

Figure 9: Model Architecture

### C.2 DISCUSSION FOR INPUT FORM

The decoder in NLP task originally sets the *shifted output* as input. Take the German-to-English translation scenario as an example where the embedded word vectors of German are set as the *Encoder* input, the model will first send a [GO] vector into the decoder and generate the first word vector of the translated English sequence, then the predicted vector will be sent into the decoder to predict the next word vector and the decoder will complete the sentence translation by repeating this operation until the end.

Mapping directly from this model setting in NLP task to our series forecasting scenario, we can also use the shifted ground truth as the decoder input, i.e., to forecast the speed of the next $T$ time stamps given the speed of the first $T$ time stamps, the data of $T \leq t \leq 2T - 1$ are sent into the decoder. Consequently, the **Casual Mask** in Vaswani et al. (2017) need to be modified to make sure that the leftward information flow is prevented.

For data forecasting by CDSA in the *Decomposed*, the masking on *Attention Map* on *Time* is simply masking out (setting to $-\infty$) the values in the input of Softmax which corresponds to the leftward information flow. Same with Vaswani et al. (2017), the masking is only adopted in the Multi-head CDSA layer labeled as (Mask TLM) in Fig. 9. However, as shown in Fig. 10, to calculate the *Attention Map* of *Location* and *Measurement* for data forecasting at $2T - 1$, all illegal units of input corresponding to $t \geq 2T - 1$ have to be masked out (setting as 0). Then, the *Masked Input* are mapped to *Query*, *Key* and *Value* to build the *Attention Map* and calculate the *Updated Value*.

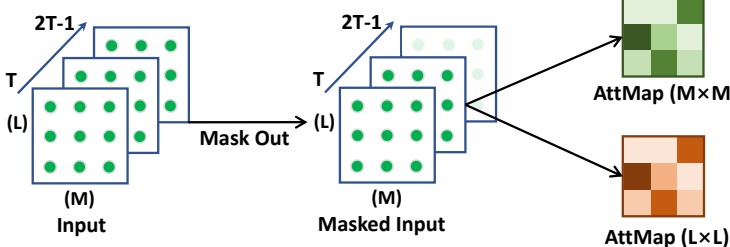

Figure 10: Casual Mask Design for Attention Map on *Location*

Besides, the decoder generates predictions given previous ground truth observations during training while the ground truth observations are replaced by predictions generated by the model itself during testing. As, the discrepancy between the input distributions of training and testing can cause degraded performance, We adopt the *integrated sampling* Bengio et al. (2015) as in Li et al. (2018) to mitigate this impact while this method is very time-consuming for the *Transformer* framework. During testing,

Table 10: Comparisons of Prediction Performance on dataset **METR-LA**

| Time | Metric | Shifted Output (Mean Final) | Shifted Output (Mean Step) | Encoder Input (Complemented) | Encoder Input (Mean) |
|---|---|---|---|---|---|
| 15 min | MAE | 3.05 | 3.09 | 3.15 | **3.01** |
| | RMSE | 5.46 | 5.50 | 5.31 | **5.08** |
| | MAPE | 8.31% | 8.47% | 8.18% | **7.82%** |
| 30 min | MAE | 3.49 | 3.54 | 3.47 | **3.14** |
| | RMSE | 6.62 | 6.66 | 5.96 | **5.38** |
| | MAPE | 9.56% | 9.88% | 9.19% | **8.30%** |
| 60 min | MAE | 4.21 | 4.26 | 4.17 | **3.40** |
| | RMSE | 8.14 | 8.19 | 7.67 | **6.27** |
| | MAPE | 11.44% | 11.79% | 11.94% | **9.76%** |
| Mean | MAE | 3.51 | 3.56 | 3.54 | **3.16** |
| | RMSE | 6.56 | 6.61 | 6.14 | **5.48** |
| | MAPE | 9.56% | 9.84% | 9.53% | **8.50%** |

In summary, by setting shifted output as the **Decoder** input, multiple *Attention Map* are calculated for forecasting value of different time stamps which requires huge memory usage. Still, integrated

sampling makes this framework suffer from an exhausted training time, since we need to send the predicted output back to decoder (*Run*) and repeat this *Run* for $T$ times. During testing, we can use the output corresponding to its own *Run* (*Step*) as the predicted result, as well as the output of the last run (*Final*). As shown in the first 2 columns in Table 10, the performance of outputs in the last run (*Final*) is better than that of *Step* mode, which means the leftward information flow still exists to break the auto-regressive property in data forecasting even though the mask is adopted on the input data. For fair comparison, the models for testing are trained in one GPU and the training time are all less than 50 hours.

Typically, missing value still exists in the original dataset. During experiment, we use the global mean to replace the missing value (*Mean*). We also compare the prediction performance between the input with *Mean Filling* and *Complemented Input* of Sec. 3.3. and the results in Table 10 shows the *Complemented Input* does not lead to performance improvement but increase the training workload. Consequently, we make encoder and decoder share the same input to reduce the memory usage and training time while our model achieves better performance for long-term prediction.

# D DATASET DESCRIPTION

## D.1 DATA AGGREGATION ON **KDD 2018** DATASET

Table 11: Selection of Common Locations between *Air Quality* and *Meteorology*

|    | *Air Quality* | *Meteorology* |
|----|---------------|---------------|
| 1  | fengtaihuayuan | fengtai_meo |
| 2  | fangshan | fangshan_meo |
| 3  | daxing | daxing_meo |
| 4  | tongzhou | tongzhou_meo |
| 5  | shunyi | shunyi_meo |
| 6  | pingchang | pingchang_meo |
| 7  | mentougou | mentougou_meo |
| 8  | pinggu | pinggu_meo |
| 9  | huairou | huairou_meo |
| 10 | miyun | miyun_meo |
| 11 | yanqin | yanqing_meo |

The original **KDD 2018** dataset consists of an Air Quality data of 35 locations and an Meteorology dataset of 18 locations. each dataset contains 6 different measurements. During experiment, Luo et al. (2018a) select 11 common locations between the two datasets and the measurements of paired locations are concatenated. The location pairs are described in Table 11. Since the unit of some measurements are label-based, e.g., the measurement *weather* denotes the types of weather including sunny, rainy and etc, these label are replaced with value such as $1, 2, ..., 9$. As the range of different measurements varies, Luo et al. (2018a) first apply *Z-score* normalization for each measurement and the MSE is calculated based on normalized data while the metrics calculation for other dataset are based on the original data.

## D.2 NYC TRAFFIC DATASET

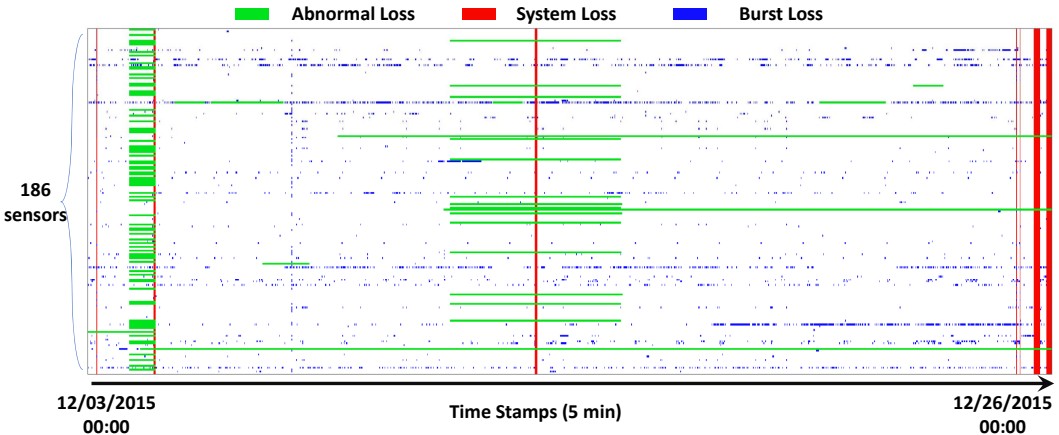

Figure 11: Loss condition of **NYC** dataset: The horizontal axis represents time line while the vertical axis represents sensors. Each unit indicates whether the data is missed in a 5-min window. The white area indicates available observation. The blue area indicates the *Burst Loss*. The red area indicates the time slots when the data of all the sensors are missed. The green area denotes as *Abnormal* for a certain location the data is continuously missed for a very long period.

**Traffic volume extraction:** We extract the traffic volume from images using a Faster R-CNN with VGG16 backbone, trained on the MIO-TCD Luo et al. (2018b) dataset. The dataset contains 110k training images, with bounding box annotation for 11 vehicle categories (articulated truck, bicycle, bus, car, motorcycle, motorized vehicle, non-motorized vehicle, pedestrian, pick-up truck, single-unit truck, work van). On the NYC-traffic dataset, we manually annotated bounding boxes for a portion of images to evaluate the vehicle detector. Our model achieves 73% precision and 54% recall with an IoU of 0.5. To construct the **NYC** traffic time series, we use the model to extract and then sum up the #cars of 11 different types (articulated truck, bicycle, bus, car, motorcycle, motorized vehicle, non-motorized vehicle, pedestrian, pick-up truck, single-unit truck, work van), in non-overlapping 5-minute intervals.

**Burst loss simulation:** As shown in Fig. 11, we term the loss area marked as blue as the *Burst loss* area where for a certain camera, the data is continuously missed for a $\delta$ time slots. After statistics and analysis, we found the the length of time slots $2 \leq \delta \leq 134$. Then, for those time slots of burst loss, we calculate the mean $\mu = 6.350773$ and standard deviation $\delta = 9.809643$. With the mean and standard deviation, We model the generation of burst loss as Gaussian process.

# E    DISCUSSION

## E.1    VISUALIZATION OF DIMENSION-SPECIFIC ATTENTION MAP

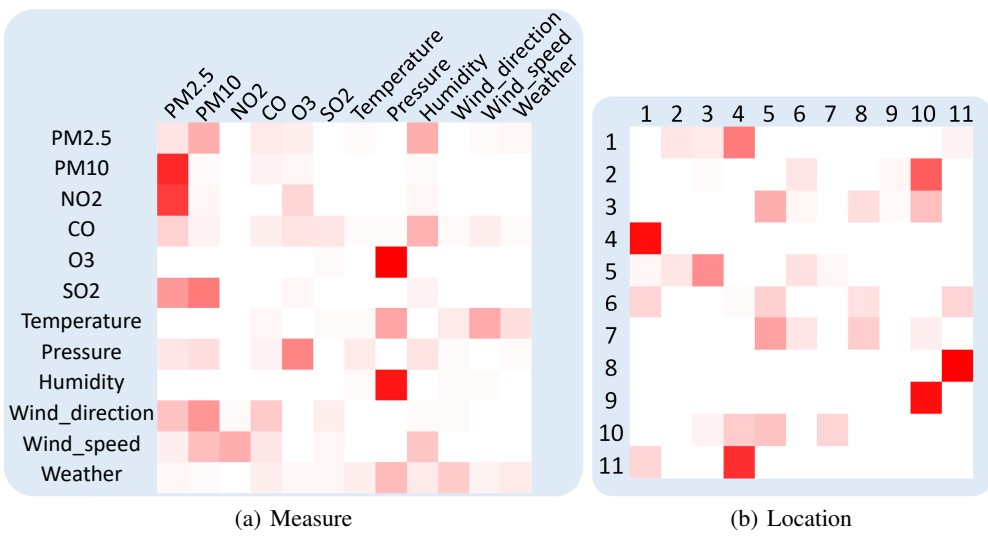

(a) Measure                                    (b) Location

Figure 12: Attention map example of the last CDSA layer

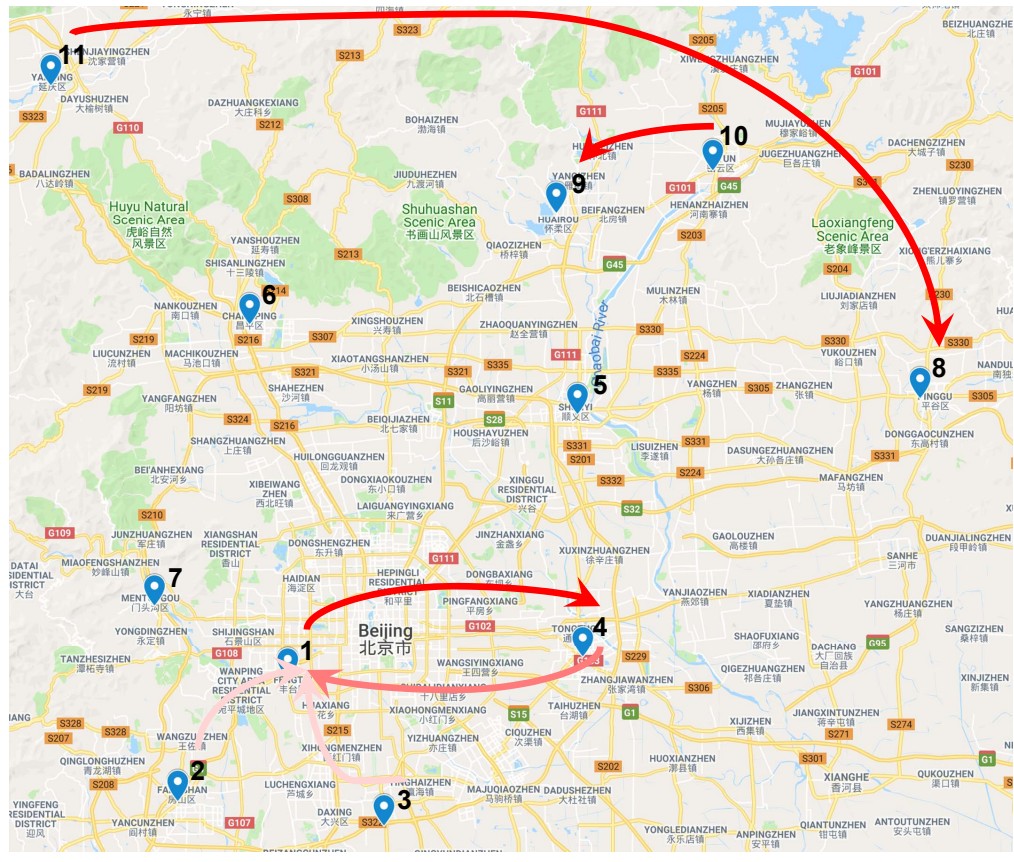

Figure 13: **KDD-2015** Visualization of Location Correlation (arrow with darker color indicates a higher weight)

We provide the attention map examples extracted from the last CDSA layer. As shown in Fig. 4(b), correlation exists between different measurement, i.e., PM2.5 and PM10 are highly correlated. As shown in Fig. 12(a), the estimation of PM2.5 and PM10 is also highly relied on each other, i.e., for the estimation of PM2.5, the color in second unit, representing the weight of PM10, is darker than the rest in the first row.

As shown in Fig. 12(b) and Fig. 13, the arrow/unit with deeper color indicates a higher weight and the index of location can be found in Table 11. According to the map in Fig. 13, in most cases, neighbouring locations often share higher attention weight, e.g., the estimation at location 1 is mainly relied on the available data from location 2, location 3 and location 4. However, the estimation of location 11 is not relied on its neighbor (location 6), instead, it is mainly relied on the location 8. We think this relation is induced since both location 11 and location 8 are the center of express way while they are away from the urban area. Thus, the air condition from those two location my highly correlated.

### E.2 Visualization of Attention Map for Cross-dimensional Imputation

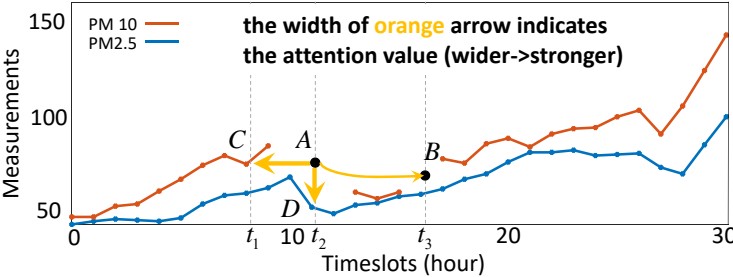

Figure 14: Visualization of prediction of missing point A, our model not only attends to available points (e.g. C, D) but also attends to missing points (e.g. B).

Besides the sample in Fig. 4 where the missing value can be estimated from the cross-dimensional available data, Fig. 14 visualizes another example and further shows that when predicting missing value A, our model pays strong attention to available values C and D while also some attention to another missing value B.

## E.3 RUNNING TIME COMPARISON

Table 12: Computational complexity of several methods to implement CDSA

| Methods | Independent | Joint | Shared | Decomposed |
|---|---|---|---|---|
| Average running time / sample (ms) | 218 | 859 | 159 | 161 |

Following the model hyper-parameter setting in Tabls. 1, we further compare the average running time for one segment during testing. As the way of attention modeling determines the computational efficiency, computation method with higher FLOPs also leads to longer running time. As shown in Table. 12, the running time of *Joint* is much higher than the rest 3 methods. Since the computation schemes of *Shared* is similar with the *Decomposed*, while the number of trainable variables of *Shared* is much less that of *Decomposed*, the average processing time of *Shared* is a bit smaller than the running time of the *Decomposed*.

## E.4 DETAIL OF FORECASTING BY USING THE IMPUTED TIME SERIES

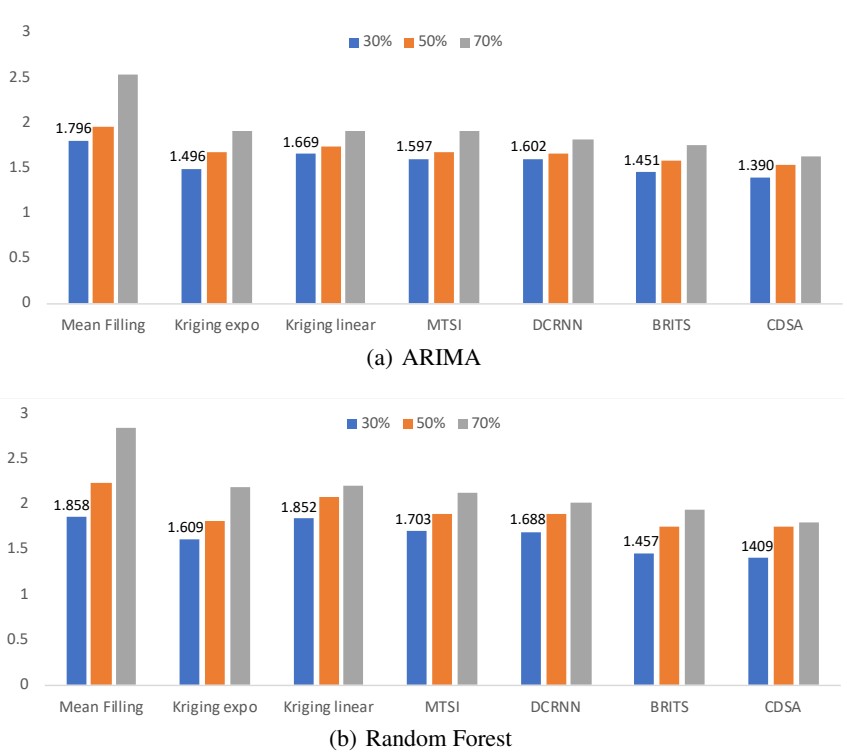

(a) ARIMA

(b) Random Forest

Figure 15: RMSE comparison for Downstream Forecasting on **NYC-Traffic**

As described in the main paper, we use the 23-day data of **NYC-Traffic** for further forecasting. We split the data into two segments, one segment contained the data of the first 20 days and the other contained the data of the rest 3 days. We used the imputed data from the first segment to forecast the value of the second segment. To provide comprehensive comparison, according to different missing ratio (i.e., 30%, 50% 70%), we remove the value of some units in the first segment according to burst loss and then feed the segment with missing value into the data statistical imputation model (i.e., Mean filling, Kriging expo, Kriging linear) and deep learning methods (MTSI, DCRNN, BRITS, CDSA(ours)). Then, we feed the imputed data in to prediction model (ARIMA, Random Forest Friedman et al. (2001)) and evaluate the forecasting performance in terms of RMSE. According to the Fig.15, we can find our proposed model always outperforms than other method.

## F  EXPERIMENT SETTING

### F.1  BASELINES

We compare our method with both deep learning based methods and statistical methods while all statistical methods are adopted for each measurement individually.

- **Mean Filling**: Replace the missing data with global mean.
- **Auto Regressive** Akaike (1969); Orfanidis (1988); Haddadi et al. (1998): Aggregate both forward and backward auto regressive on each time series data by weighted average and replace the missed value, implemented in MATLAB.
- **Kriging**: Fit linear (Exponential) function between data variance and geodesic distance and replace the missing value w.r.t. the geo-location. Applied on each time slot and implemented in PyKrige[2].
- **Multi-Imputation by Chained Equation (MICE)** Buuren & Groothuis-Oudshoorn (2010): Replace missing data by creating multiple imputations with chained equations.
- **$k$-nearest neighbor (KNN)** Speed (2003); Hastie et al. (1999); Troyanskaya et al. (2001): Use normalized Euclidean distance to find similar samples and impute the missing values with weighted average of its neighbors.
- **Matrix Factorization (MF)** Yu et al. (2016): Factorizes the incomplete matrix into two low-rank matrices and fill the missing values by $l_1$ sparsity and $l_2$ penalty.
- **MTSI** Luo et al. (2018a): Treat each measurement at different location as variables identically, align the time series of all the variables into one matrix and impute the missed data through RNN-based GAN.
- **ST-MVL** Yi et al. (2016): Replace the missed value by using geo-location information
- **BRITS** Cao et al. (2018): Treat the missing data as variable of the bidirectional RNN and impute by getting the *delayed* gradients for missing values in both forward and backward directions.
- **IIN** Zhou & Huang (2018): Connect three bi-directional RNN in turn while the output of each RNN are adopted in loss calculation.
- **FC-LSTM** Sutskever et al. (2014): Apply fully connected LSTM hidden units in Encoder-Decoder RNN structure to model the temporal dependency and forecast the traffic speed.
- **DCRNN** Li et al. (2018): Model traffic speed as signals diffused over bidirectional graph and modify the gated recurrent unit in encoder-decoder RNN structure to capture the spatiotemporal dependency.
- **DST-GCNN** Wang et al. (2018): Build a two-stream framework for model the *condition* stream and *flow* evolution while the model use the predicted *condition* to forecast the *flow*.
- **GaAN** Zhang et al. (2018b): Use attention mechanism for each location and then graph aggregator to assemble the neighbor nodes and impute the missing value.

---

[2]https://github.com/bsmurphy/PyKrige

## F.2 IMPLEMENTATION DETAILS

Table 13: Statistics of Dataset

| Dataset | Task | # location | # meas. | duration | Start | End |
|---|---|---|---|---|---|---|
| **KDD-2018** | Imputation | 36 | 12 | 364 days | 1/1/2017 | 12/30/2017 |
| **NYC Traffic** | Imputation | 186 | 1 | 23 days | 12/03/2015 | 12/26/2015 |
| **METR-LA** | Forecasting | 207 | 1 | 122 days | 3/1/2012 | 6/30/2012 |
| Dataset | #Training | #Testing | #Validation | unit duration | seg. length | Loss function |
| **KDD-2018** | 182 | - | 182 | 2 hours | 48 hours | RMSE |
| **NYC Traffic** | 173 | - | 173 | 5 minutes | 36 hours | RMSE |
| **METR-LA** | 23991 | 3404 | 6831 | 5 minutes | 1 hour | RMSE |

The dataset in **METR-LA** also has missing data while the missing rate is 91%. Thus, the segment sample whose all units are zero, i.e., all-zero sample, exists. During training, the all-zero sample (in training set) essentially has no contribution for the model training. During testing and validation, the evaluation metric will of such samples will not be counted.

**Data Pre-processing** We apply *Z-score* normalization on each measurement as (9) respectively and fill the missing value with 0.

$$
\begin{aligned}
\boldsymbol{X}(t,l)' &= \frac{\boldsymbol{X}(t,l) - \mu}{\sigma}, \\
\mu &= \frac{\sum_{t=1}^{T}\sum_{l=1}^{L}\boldsymbol{X}(t,l)}{TL}, \\
\phi &= \sqrt{\frac{\sum_{t=1}^{T}\sum_{l=1}^{L}(\boldsymbol{X}(t,l) - \mu)^2}{TL}}.
\end{aligned}
\tag{9}
$$

**Optimizer** We use the Adam optimizer Kingma & Ba (2014) while the initial learning rate in each epoch is set as

$$
lr(e) = r_0 \times \alpha^{ceil(\max(0, e-d)/i)}.
\tag{10}
$$

## F.3 KDD-2015

For **KDD-2018**, Luo et al. (2018a) adopts *content loss* in a GAN-based model to train the random noise and then estimate the missing value, i.e., for one data segment, according to the specified missing rate, some available data will be held to evaluate the imputation performance, while the remaining available data will be set as groundtruth to calculate the *content loss*. Our experiment on **NYC-Traffic** follows the same experiment setup as **KDD-2018** while the noise is replaced with the remaining available data and the model parameter is trained according to the content loss. Thus, there is no division of training, validation or testing since the training loss is not calculated from the held available data.

To comprehensively develop our experiment, we also adopting our method on **KDD-2015** and follow the experiment setup in Yi et al. (2016); Cao et al. (2018) while the available data will be trained to predict the held data directly.

**KDD-2015** (Zheng et al. (2015)). This dataset focuses on air quality and meteorology. It contains data recorded hourly, ending up with totally 8,759 time stamps. PM2.5 measurement is recorded at 36 locations and Temperature and Humidity are recorded at 16 locations in Beijing from 05/01/2014 to 04/30/2015, with natural missing rate 13.3%, 21.1% and 28.9% respectively. We treat those two subsets as two separate tasks and evaluate our method on each task separately. Following Yi et al. (2016), data in $3^{rd}, 6^{th}, 9^{th}$ and $12^{th}$ months are for testing and the remaining months are for training. We randomly select 36 consecutive time slots to train our model and evaluate Mean Absolute Error (MAE) as well as Mean Relative Error (MRE).

In order to simulate experiments for imputation, besides the natural missing data, for PM2.5 we follow the strategy used in (Yi et al. (2016); Cao et al. (2018); Zhou & Huang (2018)) to further manually remove entries and hold the corresponding value as ground truth. The imputation task is to predict values of these manually removed entries. For Temperature and Humidity, we follow Zhou & Huang (2018) to randomly hold 20% of available data.

**KDD-2015**. Table 14 shows that for PM2.5, our method outperforms the traditional methods significantly and achieves comparable MAE as IIN (Zhou & Huang (2018)) while better MRE than IIN (Zhou & Huang (2018)). For Temperature and Humidity, our method consistently outperforms state-of-the-art methods.

Table 14: MAE/MRE on dataset **KDD-2015** for comparisons with SOTA

| Model \ Dataset | PM2.5 | | TEMP | | HUM | |
|---|---|---|---|---|---|---|
| | MAE | MRE | MAE | MRE | MAE | MRE |
| Mean | 55.51 | 77.97% | 9.21 | 97.56% | 20.34 | 57.85% |
| KNN | 29.79 | 41.85% | 1.26 | 19.83% | 7.28 | 16.22% |
| MICE | 27.42 | 38.52% | 1.23 | 18.29% | 6.97 | 15.87% |
| ST-MVL Yi et al. (2016) | 12.12 | 17.40% | 0.68 | 4.59% | 3.37 | 5.91% |
| MTSI Luo et al. (2018a) | 13.34 | 18.01% | 0.71 | 4.67% | 3.51 | 6.21% |
| BRITS Cao et al. (2018) | 11.56 | 16.65% | 0.63 | 4.16% | 3.31 | 5.68% |
| DCRNN Li et al. (2018) | 12.33 | 17.82% | 0.69 | 4.59% | 2.95 | 5.12% |
| IIN Zhou & Huang (2018) | **10.63** | 15.31% | 0.63 | 4.22% | 2.90 | 5.09% |
| **CDSA (ours)** | 10.67 | **14.89**% | **0.61** | **4.15**% | **2.81** | **4.92**% |

Table 15: Comparisons of different manners to implement CDSA on dataset **KDD-2015**.

| Model \ Dataset | PM2.5 | | TEMP | | HUM | |
|---|---|---|---|---|---|---|
| | MAE | MRE | MAE | MRE | MAE | MRE |
| CDSA(Independent) | 11.54 | 16.01% | 0.68 | 4.40% | 3.19 | 5.42% |
| CDSA(Joint) | 11.20 | 15.52% | 0.65 | 4.27% | 3.05 | 5.37% |
| CDSA(Shared) | 13.85 | 19.26% | 0.75 | 5.18% | 3.56 | 6.47% |
| **CDSA(Decomposed)** | **10.67** | **14.89**% | **0.61** | **4.15**% | **2.81** | **4.92**% |

Since the *Decomposed* draws attention maps as the *Independent* but shares *Value* as the *Joint*, it reduces the computational complexity significantly. As shown in Table 15, we also evaluate these methods on **KDD-2015** datasets and the *Decomposed* achieves the best performance.

F.4  METRICS

Suppose $\boldsymbol{x} = [x_1, x_2, ...x_N] \in \mathcal{R}^N$ represents the ground truth and $\hat{\boldsymbol{x}} = [\hat{x}_1, \hat{x}_2, ...\hat{x}_N] \in \mathcal{R}^N$ represents the predicted value.

Root Mean Square Error (RMSE)

$$\text{RMSE}(\boldsymbol{x}, \hat{\boldsymbol{x}}) = \sqrt{\frac{1}{N} \sum_{n=1}^{N} (x_n - \hat{x}_n)^2}$$

Mean Squared Error (MSE)

$$\text{MSE}(\boldsymbol{x}, \hat{\boldsymbol{x}}) = \frac{1}{N} \sum_{n=1}^{N} (x_n - \hat{x}_n)^2$$

Mean Absolute Percentage Error (MAPE)

$$\text{MAPE}(\boldsymbol{x}, \hat{\boldsymbol{x}}) = \frac{1}{N} \sum_{n=1}^{N} \left| \frac{x_n - \hat{x}_n}{x_n} \right|$$

Mean Absolute Error (MAE)

$$\text{MAE}(\boldsymbol{x}, \hat{\boldsymbol{x}}) = \frac{1}{N} \sum_{n=1}^{N} |x_n - \hat{x}_n|$$

Mean Relative Error (MRE)

$$\text{MRE}(\boldsymbol{x}, \hat{\boldsymbol{x}}) = \frac{\sum_{n=1}^{N} |x_n - \hat{x}_n|}{\sum_{n=1}^{N} x_n}$$

F.5  MODEL HYPER-PARAMETER

Since there are missing data (*Naturally missing data*) in the original dataset, to evaluate the model performance, we manually remove some of the available observation (*Manually removed data*) and hold those entries' value as ground truth for evaluation. The rest data are termed as *Available data*. Thus, as a counterpart of *Naturally missing data* in the original dataset, the *Naturally available data* consists of *Manually removed data* and *Available data*.

**KDD 2015**: Following the setting in Yi et al. (2016), we split the data into training set and testing set. During training, we send the *Available data* into the model to estimate the missing data while the loss is calculated based on the *Manually removed data*. During testing, we send the *Available data* into the model and the metric is calculated based on the *Manually removed data*. For model structure on Air Quality, we set $(d_L, d_T, v)$ as $(6, 12, 3)$ and 12 heads in each layer. We set $(r_0, \alpha, d, \text{batch size})$ as $(0.003, 0.2, 60, 23)$ for at most 100 epoch. The model hyper-parameter on Meteorology is same with that for Air quality except for $d_M = 15$.

**KDD-2018**: Following the setting in Luo et al. (2018a), we don't split the data for testing or validation while the imputation task assumes the completed data has no label. Thus, we send the *Available data* into the model while the loss is calculated based on *Available data* while the evaluation metric is calculated based on the *Manually removed data*. For the model structure, we set $(d_T, d_L, d_M, v)$ as $(30, 6, 14, 3)$ and 12 heads in each layer. We set $(r_0, \alpha, d, \text{batch size})$ as $(0.003, 0.2, 60, 20)$ for at most 100 epoch.

**NYC**: Like **KDD-2018**, the imputation task assumes the completed data has no label. For the estimated data, the loss is calculated on the *Available data* part while the evaluation metric is calculated on the *Manually removed data* part.

**METR-LA**: Following the setting in Li et al. (2018), we split the data into training set and testing set where this prediction task assumes the predicted data has labels. For model structure, we set $(d_T, d_L, v)$ as $(14, 6, 3)$ and there are 16 heads in each layer. During training, we set $(r_0, \alpha, d, \text{batch size})$ as $(0.008, 0.5, 40, 16)$.

