# OpenReview forum: "Cross-Dimensional Self-Attention for Multivariate, Geo-tagged Time Series Imputation"
_ICLR.cc/2020/Conference — Reject_

### Official Review · AnonReviewer1 · 2019-10-20
**Official Blind Review #1**

**Rating:** 6

**Review:**

The classic transformer network is designed to capture the dependencies along one dimension. Applying the transformer to spatiotemporal data creates the challenge of modeling the attention in a computationally efficient way (time, location, variable). This paper investigates different ways of implementing 3D attention and extensively evaluates the performance of them.

The proposed problem is a practical problem and I think the main contribution of this paper is valuable. However, there is always a trade-off between having access to the entire history and the ability to be used in streaming settings. The need for the processing of the entire history of time series increases the latency of the algorithm. Also, the authors should report the actual run-time of the algorithms on the real data (beyond Table 1 and compared to the baselines).

The main criticism of the experiments is that the datasets are very small. For example, NYC-Traffic has only 186 time series which is considered to be of the toy-scale. Also, if you have run the experiments and not copied the reported data, make sure to indicate that in the paper. Citing a paper in a table usually means the numbers are reported from the paper.

The ways that the authors decompose the attention tensor reminds me of the works in different tensor decomposition algorithms in spatiotemporal data [1, 2].

There are minor typos in the paper too. For example, see the first in-line equation in Section 3.1.

[1] Kolda, T. G., & Bader, B. W. (2009). Tensor decompositions and applications. SIAM Review, 51(3), 455-500.
[2] Bahadori, M. T., Yu, Q. R., & Liu, Y. (2014). Fast multivariate spatiotemporal analysis via low-rank tensor learning. In NeurIPS.


**Experience Assessment:**

I have published one or two papers in this area.

**Review Assessment: Checking Correctness Of Derivations And Theory:**

I assessed the sensibility of the derivations and theory.

**Review Assessment: Checking Correctness Of Experiments:**

I carefully checked the experiments.

**Review Assessment: Thoroughness In Paper Reading:**

I read the paper at least twice and used my best judgement in assessing the paper.

---

> ### Author Response · Authors · 2019-11-15
> **Response to Review #1**
>
> We appreciate that R1 acknowledges “problem is practical” and our contribution is “valuable” .
>
> Q1: "the size of dataset in experiment is small".
>  The KDD-2018 and METR-LA are two benchmarks used in recent papers. For the NYC-Traffic, the number of locations (#location) is 186 and is comparable with #location, 207, in METR-LA. For research purpose, we gather a 2-month data of snapshots from all 186 street cameras in Manhattan and the size of the snapshot dataset is about 5TB. We apply the faster-RCNN vehicle detection model to detect the number of vehicles of the whole dataset and then calculate the traffic volume of each snapshot to build the NYC-Traffic dataset. We put month of effort to train the model and improve the detection accuracy. We are willing to put much more effort to apply for the access of snapshots in longer period and conduct research to build a larger dataset.
>
> Q2: Paper citing in experiment results.
> On NYC-Traffic, the experiment results on baseline models are done by ourselves. We appreciate you constructive suggestion about the citing rule in table and we will revise the paper accordingly as you instructed.
>
> Q3: The ways of CDSA is similar to tensor decomposition.
> Actually, the demonstration of CDSA order-independent property in based on the Kronecker product, which is an operation defined in tensor domain. We are willing to study more about tensor decomposition and provide discussion on [1,2] in the final paper.
>
> Q4: "running time comparison".
> Our model is designed for the imputation of archived data. However, we do not need to acquire the data of the entire history for data imputation. Instead, we only need to use the data of 36 hours (in NYC-Traffic) and 2 days (in KDD-2018) and 1 hour (in METR-LA).  The dataset used for evaluation in Table 1 is KDD-2018 and we have extended Table 1 in paper to Table 12 in Supp. Sec. E.3 by providing the average running time of each time series sample. As requested, we provide the average running time on baseline and CDSA in Table M1 and we will provide the running time comparison on other dataset in the final paper.
>
> Table M1: Running time Comparison
> =========================================================================
> Methods                  Independent    Joint   Shared   Decomposed   BRITS   MTSI  DCRNN
> Average time(ms)        218               859        159              161              722       803      737  =========================================================================

---

### Official Review · AnonReviewer3 · 2019-10-23
**Official Blind Review #3**

**Rating:** 1

**Review:**

This paper empirically studies the effectiveness of transformer models for time series data imputation.  In particular, the paper studies the effect of generalized forms of self-attention that can attend across dimensions of the input.

Generalizing self attention to work across dimensions of a multi dimensional time series is a good idea, and the experiments in the paper seem to support its effectiveness.  The paper does provide some ablation results to compare their three forms of modified self attention, which is good.  I believe the primary contribution of the paper is as an empirical study into the effectiveness of generalized self attention in time series datasets.

However, I have to vote to reject the paper.

My primary issue with the paper is its tone.  While generalized forms of self attention are a good idea, the paper strongly emphasizes that it is a novel idea.  In particular, image models that use self attention regularly attend too multiple dimensions.  Consider for instance arXiv:1802.05751, arXiv:1904.09793, arXiv:1712.09763, arXiv:1805.08318.

There is also several existing generalized self attention discussions: arXiv:1812.01243 or arXiv:1805.00912

The idea of extending self attention to look at multiple dimensions is fairly obvious.  If the paper changed its tone from purporting to construct a novel method of self-attention (which I do not believe it does), to being an empirical study of the utility of self attention models for doing time series imputation I would be much more willing to accept it, though as a purely empirical study the bar would be high on the standard of the experiments, the reported experiments are on rather small datasets.

Also, the paper could use an edit for grammar.

**Experience Assessment:**

I do not know much about this area.

**Review Assessment: Checking Correctness Of Derivations And Theory:**

I did not assess the derivations or theory.

**Review Assessment: Checking Correctness Of Experiments:**

I assessed the sensibility of the experiments.

**Review Assessment: Thoroughness In Paper Reading:**

I read the paper at least twice and used my best judgement in assessing the paper.

---

> ### Author Response · Authors · 2019-11-15
> **Response to Review #3**
>
>  We appreciate R3 for pointing out relevant papers about self-attention generalization and we acknowledge that a lot of works have been done to generalize the self-attention on multi-dim data. In addition to our contribution on generalizing self-attention to multi-dim data for imputation and forecasting tasks, we want to clarify the key difference between CDSA in decomposed form and previous works. In the decomposed form:
> (1) CDSA treats each dimension equally and can formulate attention maps for each dimension. In this way, we don’t need to flatten the multi-dim data into a huge vector but still can build the cross-dimensional attention map to model the global dependency.
>  (2) we provide theoretical proof to show the order-independent property when CDSA uses the dimension-specific attention maps of different dimension to modulate the value vectors in sequence.
>
> We carefully studied the papers R3 provided. We found that all algorithms in those papers generalized the self-attention by first fusing multiple dimensions into one single dimension. In detail, algorithms in (arxiv: 1812.01243, 1904.09793, 1712.09763, 1802.05751) directly model the dependency between each two samples/pixels by first flattening all samples/pixels into one single vector. The algorithm in arxiv: 1805.08318 first uses a CNN-based network to extract features from the image and flattens the features into one single dimension. The algorithms in papers (arxiv: 1709.04696, 1805.00912) are still applied on NLP tasks and the application on multi-dim data is not clearly explained.
>
> By building the dimension-specific attention for the geo-tagged, multi-variate time series data, we are able to infer the temporal relationship, spatial correlation and measurement correlation respectively. For example, in traffic speed forecasting, when a traffic jam happens at one location, the average traffic speed of the related locations are impacted. Then, according to the spatial attention map, we can infer the source location of traffic jam that impacts the traffic speed of the other locations. For air quality imputation, in Fig. 12 (right) and 13, we also show the spatial attention map and its visualization. In this way, we can infer the potentially highly-correlated locations according to the dimension-specific attention map of location. Then, we can further set those locations as reference for data imputation.
>
> In addition, our proposed CDSA has low FLoPs and is efficient in memory resource usage, which enables fast processing. Notably, different from the decomposed attention in arxiv 1812.01243, CDSA does not impose any extra constraint on the intermediate variable (i.e. query and key) when compared with self-attention under NLP[M1]. Thank you for your advice and we will revise the grammar of our paper carefully in the final paper.
>
> [M1] Vaswani A, Shazeer N, Parmar N, Uszkoreit J, Jones L, Gomez AN, Kaiser Ł, Polosukhin I. Attention is all you need. InAdvances in neural information processing systems 2017 (pp. 5998-6008).

---

### Official Review · AnonReviewer4 · 2019-11-03
**Official Blind Review #4**

**Rating:** 3

**Review:**

This paper proposes a Transformer-based model with cross-dimensional self-attention for multivariate time series imputation and forecasting. The authors consider time series with observations collected at different locations (L), for different measurements (M), and at different timestamps (T). The authors describe 4 different self-attention mechanisms based on how the three dimensions, and it turns out the proposed Decomposed approach achieves the best performance and has moderate model complexity among the four. Experiments on several traffic and air quality datasets show the superiority of the proposed model.

Overall the problem and the proposed model are well motivated. Handling time series data with missing values is quite important, and the authors design the novel cross-dimensional attention mechanism which is reasonable and performs well. Especially, the authors compare several recent RNN-based models. The proposed method outperforms these strong baselines in imputation and long-term forecasting tasks.

However, I do have a few questions and concerns.

It would be quite helpful to see the training time and model size comparisons with baselines, which is to validate the claim in the introduction that `replacing the conventional RNN-based models to speed up training`.

The proposed model treats all three dimensions equally, with direct attention of every two variables, and is independent with the order. Though effective and mathematically clean, I am not sure the temporal/spatial smoothness and dependencies in time series are properly modeled in this way -- as time series is not the same as embedded sequences in NLP. This may explain why the performance on short-term forecasting is relatively unsatisfying.

It seems that the proposed model is designed for missing completely at random (implied by the statement `... due to unexpected sensor damages or communication errors` from the introduction, and the experimental settings on adding missing values). Many missing variables in time series may be missing at random or even not at random.

About the experiments:
Two datasets of the three mentioned in the main paper have M=1, which degrade the proposed model from 3-dimensional to 2-dimensional.
Why forecasting experiment is conducted on METR-LA, while using the imputation for forecasting is conducted on a different dataset and without comparing other forecasting baselines?
What are the metrics used in Tables 6, 7, 9? Several metrics are used (RMSE, MSE, MAPE, MAE, MRE) while results on different datasets are shown in different but not all metrics. Is there any reason to cherry-pick metrics for different experiments?
In Table 4, the proposed method's results in RMSE is consistently better than shown in MAE compared with other baselines. Any explanations would be useful.

The overall idea is relatively easy to follow, while some detailed descriptions should be added or clarified.
When taking health-care data as an example of geo-tagged time series, could you explain or provide references?
Figure 2 only demonstrates 3 attention mechanisms, and the Shared should also be included.
S(i,j) is used in Section 3.1 without explicit formal definition.
Please clarify the sentence about \sigma below Equation (4).
Please refer to the section number explicitly in Supp if used. (E.g., on Pages 5 and 7.)
On which dataset and what settings are the results in Table 1 computed? The numbers are helpful, but it would be better if the results are computed based on the hyperparameters (e.g., T, L, M, d_V, etc).

Minor typos:
Page 3, Paragraph of RNN-based data imputation methods: `...indistinguishable. so that...`
Page 3, Section 3.1, `..where Then, ...`
Page 14, A' is used to denote the reshaped tensor, while \tilde is used in the main paper.
Page 16, `During testing,`

**Experience Assessment:**

I have published one or two papers in this area.

**Review Assessment: Checking Correctness Of Derivations And Theory:**

I assessed the sensibility of the derivations and theory.

**Review Assessment: Checking Correctness Of Experiments:**

I assessed the sensibility of the experiments.

**Review Assessment: Thoroughness In Paper Reading:**

I read the paper at least twice and used my best judgement in assessing the paper.

---

> ### Author Response · Authors · 2019-11-15
> **Response to Review #4 (Part 1)**
>
> We appreciate that R4 acknowledges “problem is important” and our proposed model is “well motivated and novel”. Most questions are about the model interpretation and experimental analysis. We address them below.
>
> Q1: "running time and model size comparison".
>
> Per requested, we compare the running time & model size between the RNN-based models and our proposed CDSA. The comparison results on NYC-Traffic, KDD-2018 and METR-LA are shown in Table M1, M2, M3 below.  The training time needed for convergence of CDSA is consistently less than the training time of RNN-based baselines on all three different datasets. In particular, for the imputation task on NYC-Traffic and KDD-2018, training of CDSA is significantly faster than training of the RNN-based models (5-8 times faster).
>
> The RNN-based models only output the prediction of one timestamp each time and need to process each time stamp sequentially in order to predict the entire time series. Moreover, the gradient vanishing problem will make it difficult to train an RNN-based model for long-term prediction. However, CDSA can directly output prediction for all timestamps at once and is able to model the relationship between each two timestamps. In this way, even though CDSA’s model size is larger than the size of RNN-based models, training of CDSA is still consistently faster than training of RNN-based baselines.
>
> Table M1: Comparison on NYC-Traffic Dataset
> ========================================================
> 	Model		            MTSI        DCRNN	BRITS       CDSA(ours)
> # Variables 	                  486288       393152      372477        2981002
> Training Time (min)         2241	        2116	  1937		408
> ========================================================
>
> Table M2: Comparison on KDD-2018 Dataset
> ========================================================
> 	Model		            MTSI        DCRNN	BRITS       CDSA(ours)
> # Variables 	                 1871019    1990332   1797676       2315912
> Training Time (min)         357		364	          410		52
> ========================================================
>
> Table M3: Comparison on METR-LA Dataset
> ========================================================
> 	Model		            MTSI        DCRNN	BRITS       CDSA(ours)
> # Variables 	                  565743      355096      319325        3160741
> Training Time (min)         553	         500	           480		441
> ========================================================
>
>
> Q2: The modeling of temporal/spatial smoothness and dependencies in time series forecasting.
>
> We propose CDSA to generalize the self-attention mechanism to multi-dim data without imposing additional constraint. However, CDSA is flexible and can be expanded to incorporate new spatio-temporal constraints.
>
> Currently, we normalize all spatial pairwise distances to the range [0,1] and use the cosine of the normalized distance as location embedding to indicate spatial similarity. We then add the spatial similarities to the coefficients in attention map, which are then used to modulate value vectors in the self-attention process. In this way, we induce spatial smoothness constraint since the weights of two adjacent locations in modulation are imposed to be higher than those for distant locations. In Table 4, the prediction performance is based on the model with the spatial smoothness constraint. Compared with the model without constraint, the performance is clearly improved.
>
> The distance-based embedding in spatial domain mentioned above can be easily extended to the temporal dimension to encourage temporal smoothness. However, appropriate embedding for temporal dynamics requires more thorough investigation in the future work because temporal dynamics could be complex (such as burstiness, spike, and other sudden changes), and cannot be clearly modeled in a simple distance-based similarity. Study of such spatio-temporal modeling in the context of CDSA is a very interesting direction as future extension of our proposed CDSA method.

---

> ### Author Response · Authors · 2019-11-15
> **Response to Review #4 (Part 2)**
>
>
> Q3: The missing variables in time series may be missing at random or even not at random.
>
> In the main paper and supplementary material, we provide experiment results of three different missing conditions.
>
> In the main paper and supplementary material, we provide experiment results of three different missing conditions.
> NYC-Traffic:  missing is at random but the missing pattern is modeled as burst loss.
> Burst loss assumes that data missing at different locations are independent of each other. For each location, the duration of continuous missing data is drawn from a Gaussian distribution.
> KDD-2018: data missing is at random and no specific missing pattern is modeled.
> KDD-2015 in Supp. Section F.3.: missing value is fixed. In detail, for each location and each month, the available data of one timestamp will be held as the ground truth for evaluation if data of the same timestamp in the last month is missing. This missing value selection strategy of missing pattern is introduced from ST-MVL[M1]. Still, our method outperforms the provided baseline methods.
>
> On KDD-2018 and KDD-2015,  our  model  is  able  to  learn  both  spatial/temporal  dependency  and  measurement correlation automatically and can then make accurate estimation.
>
> Q4: About experiment.
>
> Q: The measurement dimension is 1 in two of three datasets;
> A: In addition to the dataset KDD-2018, we also conduct data imputation on KDD-2015
> with 3 different measurements in Supp.  Section F.3.  CDSA outperforms baselines in most cases.  We will conduct experiments on other high-dimensional dataset in the final paper.
>
> Q: The forecasting task is on METR-LA, while using the imputation for forecasting is conducted on NYC-Traffic and without comparing other forecasting baselines?
> A: The forecasting on the imputed NYC-Traffic dataset is viewed as the downstream task of imputation. The ARIMA and Random Forest are effective algorithms for time series forecasting. As requested, we set the missing rate before imputation at 50% and extend the result in the main paper by comparing forecasting performance on DCRNN and CDSA in Table M4. We will provide more comparison results in the final paper.
>
> Table M4: Forecasting (downstream task) comparison on imputed data (50% missed)
> =========================================================================
>                 Mean Filling.  Kriging expo.  Kriging linear.  MTSI.  DCRNN.  BRITS.  CDSA(ours)
> DCRNN.       1.737                 1.523               1.647             1.634    1.516     1.342       1.271
> CDSA.           1.751	                1.517               1.621	       1.482    1.566      1.305      1.258
> =========================================================================
>
> Q: Metric Clarification and Selection:
> A: The metric for KDD-2018 evaluation in Table 7 and 9 is MSE. The metric for NYC-Traffic evaluation in Table 6 and 9 is RMSE. For NYC-Traffic evaluated in Table 9, each sample is the average traffic volume in 30 min. For NYC-Traffic evaluated in Table 2 and 6, each sample is the average traffic volume in 5 min. We are sorry for this misalignment and will update the aligned result (each sample is the average traffic volume in 5 min) shown in Table M5 in Supp. Table 9.
>
> Table M5: Performance Improvement for Imputation Mask on NYC-Traffic
>  =============================================================
> Model \ Missing Rate    20%    30%    40%     50%    60%    70%    80%     90%
> 	Masked 	                1.203   1.208   1.211   1.214   1.215   1.217   1.234   1.377
> 	No Mask  	        1.909   1.942   2.117   2.203   2.211   2.243   2.259   2.635
> =============================================================
> The evaluation metrics on KDD-2018 and METR-LA follow standards in previous works. We will provide comparison of MSE, MAE, MRE on NYC-Traffic in Supp.
>
> Q: Why "RMSE is consistently better than shown in MAE compared with other baselines in Table 4".
> A: MAE measures the average magnitude of the errors in predictions.  RMSE is a quadratic scoring rule in such measuring.  As such, RMSE gives a relatively high weight to large errors and is more useful when large errors are undesirable. For the difference between the groundtruth and prediction, when compared with other baseline models, the mean of differences is a bit larger than the baseline models (BRITS, DCRNN, DST-GCNN and GaAN).However, the variance of differences is smaller, which means the situation with large difference is far less frequent. Thus, CDSA is expected to outperform baselines in the forecasting of abrupt changes. In addition, our method outperforms most baselines in both RMSE and MAE and in particular excels in long-term forecasting.

---

> ### Author Response · Authors · 2019-11-15
> **Response to Review #4 (Part 3)**
>
>
> Q5: Clarification of Description.
>
> Q: Explain or provide reference when taking health-care data as geo-tagged time series.
> A: We haven’t found previous works that taking health care data as an example of geo-tagged time series. Instead, recent works [M2,M3] treats each individual instance as a variable of a dimension. In CDSA, we may use their age and job similarity to model the attention.
>
> Q: Definition of $S(i,j)$ in Section 3.1.
> A: $S(i,j) = q_i k_j^T / \sqrt{d}$ where query vector $q_i$ and key vector $k_j$ are of d-dim.
>
> Q: How is Table 1 calculated.
>  A: The results in Table 1 is computed on KDD-2018. Specifically, $(T,L,M,d_V) = 48,12,12,3$.
>
> We appreciate the constructive comments from R4 and we will revise the paper by providing necessary clarification.
>
> [M1] Yi X, Zheng Y, Zhang J, Li T. ST-MVL: filling missing values in geo-sensory time series data.
> [M2] Cao W, Wang D, Li J, Zhou H, Li L, Li Y. BRITS: bidirectional recurrent imputation for time series. InAdvances in Neural Information Processing Systems 2018 (pp. 6775-6785).
>  [M3] Luo Y, Cai X, Zhang Y, Xu J. Multivariate time series imputation with generative adversarial networks. InAdvances in Neural Information Processing Systems 2018 (pp. 1596-1607).

---

### Decision · Program_Chairs · 2019-12-19

**Decision:**

Reject

**Comment:**

The paper proposes a solution based on self-attention RNN to addressing the missing value in spatiotemporal data.

I myself read through the paper, followed by a discussion with the reviewers. We agree that the model is reasonable, and the results are promising. However, there is still some room for improvement:
1. The self-attention mechanism is not new. The specific way proposed in the paper is an interesting tweak of existing models, but not brand new per se. Most importantly, it is unclear if the proposed way is the optimal one and where the performance improvement comes from. As the reviewer suggested, more thorough empirical analysis should be performed for deeper insights of the model.

2. The datasets were adopted from existing work, but most of them do not have such complex models as the one proposed in the paper. Therefore, the suggestion for bigger datasets is valid.

Given the considerations above, we agree that while the paper has a lot of good materials, the current version is not ready yet. Addressing the issues above could lead to a good publication in the future.